RESEARCH COMMUNICATION

# Impaired voice processing in reward and salience circuits predicts social communication in children with autism

Daniel Arthur Abrams[1]*, Aarthi Padmanabhan[1], Tianwen Chen[1], Paola Odriozola[1], Amanda E Baker[1], John Kochalka[1], Jennifer M Phillips[1], Vinod Menon[2,3]*

[1]Department of Psychiatry and Behavioral Sciences, Stanford University School of Medicine, Stanford, United States; [2]Program in Neuroscience, Stanford University School of Medicine, Stanford, United States; [3]Department of Neurology and Neurological Sciences, Stanford University School of Medicine, Stanford, United States

**Abstract** Engaging with vocal sounds is critical for children's social-emotional learning, and children with autism spectrum disorder (ASD) often 'tune out' voices in their environment. Little is known regarding the neurobiological basis of voice processing and its link to social impairments in ASD. Here, we perform the first comprehensive brain network analysis of voice processing in children with ASD. We examined neural responses elicited by unfamiliar voices and mother's voice, a biologically salient voice for social learning, and identified a striking relationship between social communication abilities in children with ASD and activation in key structures of reward and salience processing regions. Functional connectivity between voice-selective and reward regions during voice processing predicted social communication in children with ASD and distinguished them from typically developing children. Results support the Social Motivation Theory of ASD by showing reward system deficits associated with the processing of a critical social stimulus, mother's voice, in children with ASD.
**Editorial note:** This article has been through an editorial process in which the authors decide how to respond to the issues raised during peer review. The Reviewing Editor's assessment is that minor issues remain unresolved (see decision letter).
DOI: https://doi.org/10.7554/eLife.39906.001

*For correspondence:
daa@stanford.edu (DAA);
menon@stanford.edu (VM)

**Competing interests:** The authors declare that no competing interests exist.

## Introduction

The human voice is a critical social stimulus in children's environment, and engaging with vocal sounds is important for language (*Kuhl et al., 2005a*; *Christophe et al., 1994*) and social-emotional learning (*DeCasper and Fifer, 1980*) during typical development. However, children with autism spectrum disorder (ASD) are often not responsive to voices (*Kanner, 1968*; *Harstad et al., 2016*), and it has been hypothesized that voice processing deficits contribute to pronounced social communication difficulties in ASD (*Klin, 1991*; *Kuhl et al., 2005b*; *Whitehouse and Bishop, 2008*). A special case of voice processing impairments in children with ASD is a deficit in processing mother's voice (*Klin, 1991*), a biologically salient and implicitly rewarding sound for typically developing (TD) children (*Lamb, 1981*; *Thoman et al., 1977*), which is closely associated with cognitive (*Kuhl et al., 2005a*; *Christophe et al., 1994*) and social development (*DeCasper and Fifer, 1980*; *Adams and Passman, 1979*). Compared to studies of visual face processing (*Dalton et al., 2005*; *Dawson et al., 2002*; *Dichter et al., 2012*; *Pierce et al., 2001*; *Schultz et al., 2000*), very little is known regarding the neurobiology of voice processing networks in children with ASD, which is fundamental to human communication.

It remains unknown why children with ASD often do not engage with the voices in their environment. Specifically, it is not known which aspects of voice processing are impaired in children with ASD. One possibility is that sensory deficits negatively affect voice processing and contribute to social communication deficits (*Dinstein et al., 2012*; *Markram et al., 2007*; *Russo et al., 2010*; *Marco et al., 2011*; *Leekam et al., 2007*; *Woynaroski et al., 2013*). A second possibility relates to the motivation to engage with socially relevant stimuli (*Chevallier et al., 2012*; *Dawson et al., 2004*; *Pelphrey et al., 2011*; *Clements et al., 2018*). The social motivation theory of ASD posits that impairments in representing the reward value of human vocal sounds impedes individuals with ASD from engaging with these stimuli, and contributes to social interaction difficulties (*Dawson et al., 2002*; *Chevallier et al., 2012*). While this is a prominent model for considering social communication function in ASD, there has been a dearth of compelling experimental evidence showing aberrant reward processing in response to clinically meaningful social stimuli (*Clements et al., 2018*).

An important approach for testing theories of ASD is the use of human brain imaging methods and functional circuit analyses. Behavioral studies are limited in their ability to provide details regarding the neural mechanisms underlying distinct aspects of social information processing, and systems neuroscience analyses can uncover important aspects of social information processing that may be impaired in individuals with ASD. For example, the social motivation theory posits that individuals with ASD show reduced engagement and connectivity in the mesolimbic reward system, including the ventral tegmental area (VTA), nucleus accumbens (NAc), orbitofrontal cortex (OFC), and ventromedial prefrontal cortex (vmPFC), and structures of the salience and affective processing systems, instantiated in the anterior insula and amygdala, during social processing (*Chevallier et al., 2012*).

Previous brain imaging research of voice processing in adults with ASD has supported the sensory deficit model by showing reduced regional activity in voice-selective superior temporal sulcus (STS) (*Gervais et al., 2004*; *Schelinski et al., 2016*), a core region associated with structural analysis of the human voice (*Belin et al., 2000*). However, several factors have precluded thorough tests of prominent ASD theories in the context of the neurobiology of voice processing. First, there have been few studies examining voice processing in ASD, particularly when compared to the extensive face processing literature (*Dalton et al., 2005*; *Dichter et al., 2012*; *Pierce et al., 2001*; *Schultz et al., 2000*; *Baron-Cohen et al., 1999*; *Dapretto et al., 2006*). Second, previous studies have not employed biologically salient voices (e.g. mother/caregiver), which are thought to be implicitly rewarding (*Chevallier et al., 2012*), to probe brain circuit function in children with ASD. For example, a recent study in TD children showed that, compared to unfamiliar voices, mother's voice elicits activation within voice-selective, mesolimbic reward, affective, and salience, and face-processing brain regions, and connectivity between these regions predicts social communication abilities (*Abrams et al., 2016*). Third, previous studies of voice processing have focused on group differences in brain activity between individuals with ASD and matched controls but have not examined how individual variation in social communication abilities are associated with social brain circuit function in ASD. Finally, although autism has been conceptualized as a disorder of brain connectivity (*Uddin et al., 2013a*; *Wass, 2011*), previous brain imaging studies of human voice processing in ASD have focused on regional activation profiles in voice-selective cortex (*Gervais et al., 2004*; *Schelinski et al., 2016*) and have not employed a brain networks perspective. Importantly, a brain networks approach goes beyond describing activation in circumscribed brain regions and accounts for the coordinated activity in distributed brain systems during social information processing, and would provide considerable insight into aberrancies in several critical brain systems in ASD (*Di Martino et al., 2011*; *Uddin et al., 2013b*; *von dem Hagen et al., 2013*). For example, a previous resting state fMRI study investigated intrinsic connectivity of voice-selective cortex and showed that children with ASD have reduced connectivity between voice-selective STS and key structures of the mesolimbic reward system, anterior insula, and amygdala (*Abrams et al., 2013a*). Moreover, the strength of intrinsic connectivity in this network predicted social communication abilities in children with ASD. While intrinsic network findings support the social motivation theory of ASD, a critical question remains: do results from intrinsic connectivity reflect an epiphenomenon, or is aberrant brain connectivity in voice and reward brain systems during the processing of biologically salient and clinically relevant voices a signature of social communication deficits in children with ASD?

Here, we examine social information processing in children with ASD by probing brain circuit function and connectivity in response to human vocal sounds. We examined two aspects of voice

processing: (1) unfamiliar voice processing compared to non-social auditory processing (i.e. environmental sounds) and (2) mother's voice compared to unfamiliar voice processing (*Figure 1a*). The rationale for this approach is that these two levels of social information processing may reflect distinct neural signatures in voice-selective, salience, and reward processing brain systems in children with ASD. A key aspect of our analysis was to investigate whether brain activity and connectivity in response to these vocal contrasts reflects individual differences in social communication abilities in children with ASD (*Lord et al., 2000*). A second aspect of the analysis was to build on results from a previous intrinsic connectivity study of the voice processing network in children with ASD (*Abrams et al., 2013a*) to examine whether stimulus-evoked connectivity patterns within this network during unfamiliar and mother's voice processing can reliably distinguish children with ASD from TD children and predict social communication abilities in children with ASD.

## Results

### TD vs. ASD activation differences in response to unfamiliar voices

Direct group comparisons between TD children and children with ASD in response to unfamiliar female voices show that children with ASD have reduced activity in a relatively small set of brain regions confined to lateral temporal cortex (*Figure 2A*; see *Appendix 1—table 1* for effect sizes and *Appendix 1—figure 2* for within-group results). Specifically, children with ASD show reduced activity in right hemisphere planum polare (PP), an area of auditory association cortex within the superior temporal gyrus. Within-group signal level analysis showed that TD children have greater activity for unfamiliar female voices, compared to environmental sounds, in this brain region (i.e. positive βs; see *Appendix 1—figure 3A*) while children with ASD show weaker activity for this same contrast (i.e. negative βs for unfamiliar voices compared to environmental sounds). No brain regions showed greater activity for unfamiliar female voices in the ASD, compared to the TD, group.

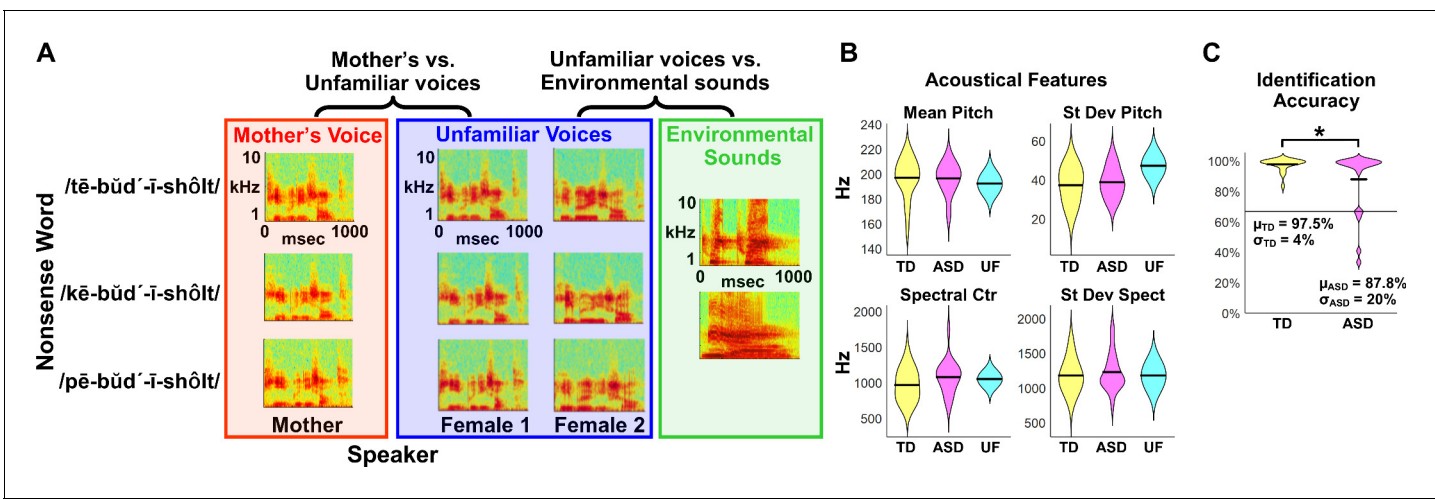

**Figure 1.** fMRI Experimental design, acoustical analysis, and behavioral results. (**A**) Randomized, rapid event-related design: During fMRI data collection, three auditory nonsense words, produced by three different speakers, were presented to the child participants at a comfortable listening level. The three speakers consisted of each child's mother and two control voices. Non-speech environmental sounds were also presented to enable baseline comparisons for the speech contrasts of interest. All auditory stimuli were 956 ms in duration and were equated for RMS amplitude. (**B**) Acoustical analyses show that vocal samples produced by the participants' mothers were comparable between TD (yellow) and ASD groups (magenta) and were similar to the control samples (cyan) for individual acoustical measures (p>0.10 for all acoustical measures; see Appendix, *Acoustical analysis of mother's voice samples*). (**C**) All TD children and the majority of children with ASD were able to identify their mother's voice with high levels of accuracy, however five children with ASD performed below chance on this measure (see Appendix, *Identification of Mother's Voice*). The horizontal line represents chance level for the mother's voice identification task.
DOI: https://doi.org/10.7554/eLife.39906.002

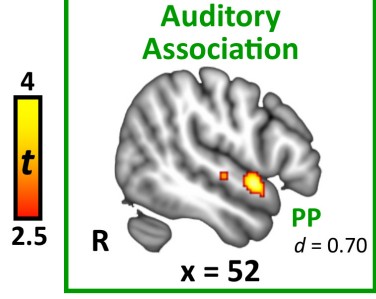

## TD Children > Children with ASD

**A** **Unfamiliar Female Voices > Non-vocal Environmental**

**B** **Mother's Voice > Unfamiliar Female Voices**

**Figure 2.** Brain activity difference in TD children compared to children with ASD in response to vocal stimuli. (**A**) Group comparisons indicate that TD children show greater activity compared to children with ASD in right-hemisphere auditory association cortex (planum polare (PP)) in response to the unfamiliar female voices > non-vocal environmental sound contrast. No regions showed greater activity in children with ASD compared to TD children for the unfamiliar female voice contrast. (**B**) Group comparisons indicate that TD children show greater activity in several visual processing regions, including bilateral intercalcarine cortex, lingual gyrus, and fusiform cortex, as well as right-hemisphere posterior hippocampus and superior parietal regions, in response to the mother's voice > unfamiliar female voices contrast. No regions showed greater activity in children with ASD compared to TD children for the mother's voice contrast.
DOI: https://doi.org/10.7554/eLife.39906.003

### TD vs. ASD activation differences in response to mother's voice

Direct group comparisons between brain responses measured from TD children and children with ASD in response to mother's voice relative to unfamiliar female voices revealed that children with ASD have reduced activity in several visual processing regions as well as key structures of the medial temporal lobe memory system (*Figure 2B*; see *Appendix 1—table 1* for effect sizes and *Appendix 1—figure 4* for within-group results). Specifically, whole-brain analysis revealed that TD children had greater activation compared to children with ASD for mother's voice in bilateral intercalcarine cortex extending into lingual gyrus. Moreover, children with ASD showed reduced activity compared to TD children in a broad extent of fusiform gyrus bilaterally, including both left-hemisphere occipital regions of fusiform as well as temporal occipital regions in the right-hemisphere. Children with ASD also showed less activity for mother's voice in right-hemisphere posterior hippocampus, a critical region for learning and memory, as well as precuneus cortex of the default mode network. Signal level analysis shows that TD children have greater activity for mother's voice compared to unfamiliar female voices in these brain regions (i.e. positive βs; see *Appendix 1—figure 3B*) while children with ASD show weaker activity for mother's voice (i.e. negative βs). No brain structures showed greater activity for mother's voice in the ASD, compared to the TD, group. Moreover, fMRI activation profiles in children with ASD were not related to mother's voice identification accuracy (see Appendix,

**Table 1.** Demographic and IQ measures

|  | ASD (n = 21) | TD (n = 21) | p-value |
| --- | --- | --- | --- |
| Gender ratio | 18 M: 3 F | 17 M: 4 F | 0.69† |
| Age (years) | 10.75 ± 1.48 | 10.32 ± 1.42 | 0.34 |
| Full-scale IQ* | 113.75 ± 15.04 | 117.45 ± 10.83 | 0.38 |
| VIQ* | 112.25 ± 16.13 | 118.55 ± 12.13 | 0.17 |
| PIQ | 111.52 ± 14.30 | 113.14 ± 13.46 | 0.71 |
| ADOS social | 9.52 ± 2.54 | - | - |
| ADI-A social | 6.81 ± 4.52 | - | - |
| ADI-B communication | 7.43 ± 5.01 | - | - |
| ADI- C repetitive behaviors | 4.10 ± 2.66 | - | - |
| Word reading | 112.24 ± 11.34 | 114.38 ± 8.96 | 0.50 |
| Reading comprehension | 108.29 ± 11.81 | 115.38 ± 9.09 | 0.35 |
| Max. Motion (mm) | 1.99 ± 0.93 | 1.73 ± 0.93 | 0.36 |
| Mother's voice ID accuracy | 0.88 ± 0.21 | 0.98 ± 0.04 | 0.04 |

Demographic and mean IQ scores are shown for the sample.

M, Male; F, Female; WASI, Wechsler Abbreviated Scale of Intelligence.

†Chi-squared test.

*Score missing for one participant in TD and ASD groups.

DOI: https://doi.org/10.7554/eLife.39906.007

fMRI activation and connectivity profiles in children with ASD are not related to mother's voice identification accuracy).

## Brain activity and social communication abilities

Identifying sources of variance in key symptom domains represents an important question for autism research. We performed a whole-brain linear regression analysis using individual social communication scores as a predictor of brain activation. We first examined this relation in the context of general vocal processing using the unfamiliar female voices minus environmental sounds contrast. Results from this analysis show a striking pattern: the strength of activity in a variety of brain systems serving auditory, reward, and salience detection is correlated with social communication abilities in children with ASD (*Figure 3A*; see *Appendix 1—table 2* for effect sizes). Specifically, this pattern was apparent in auditory association cortex of the superior temporal plane, including the PP, but also in the nucleus accumbens of the reward pathway, and anterior insula of the salience network. Scatterplots show that brain activity and social communication abilities vary across a range of values and greater social function, reflected by lower social communication scores, is associated with greater brain activity in these auditory, reward, and salience processing regions. Support vector regression (SVR) analysis (*Abrams et al., 2016*; *Cohen et al., 2010*) showed that the strength of activity in these regions was a reliable predictor of social communication function in these children ($R \geq 0.49$; $p \leq 0.011$ for all regions).

We next examined the question of heterogeneity in the context of mother's voice processing, and results show a similar pattern: children with ASD with greater social communication abilities showed greater activation for mother's voice in a wide extent of primary auditory, auditory association, and voice-selective cortex as well as mesolimbic reward, salience detection, and motor regions (*Figure 3B*). Specifically, this brain-behavior relationship was evident in auditory regions of superior temporal cortex, including medial aspects of bilateral Heschl's gyrus, which contains primary auditory cortex, right-hemisphere PP of the superior temporal plane, as well as bilateral voice-selective mSTS. This relationship was also observed in regions of the salience network, including dorsal aspects of AI bilaterally and right-hemisphere rostral ACC (rACC), as well as vmPFC of the reward network. SVR results indicated that the strength of activity in these particular brain regions during mother's voice processing was a reliable predictor of social communication function in these children ($R \geq 0.50$; $p \leq 0.009$ for all regions).

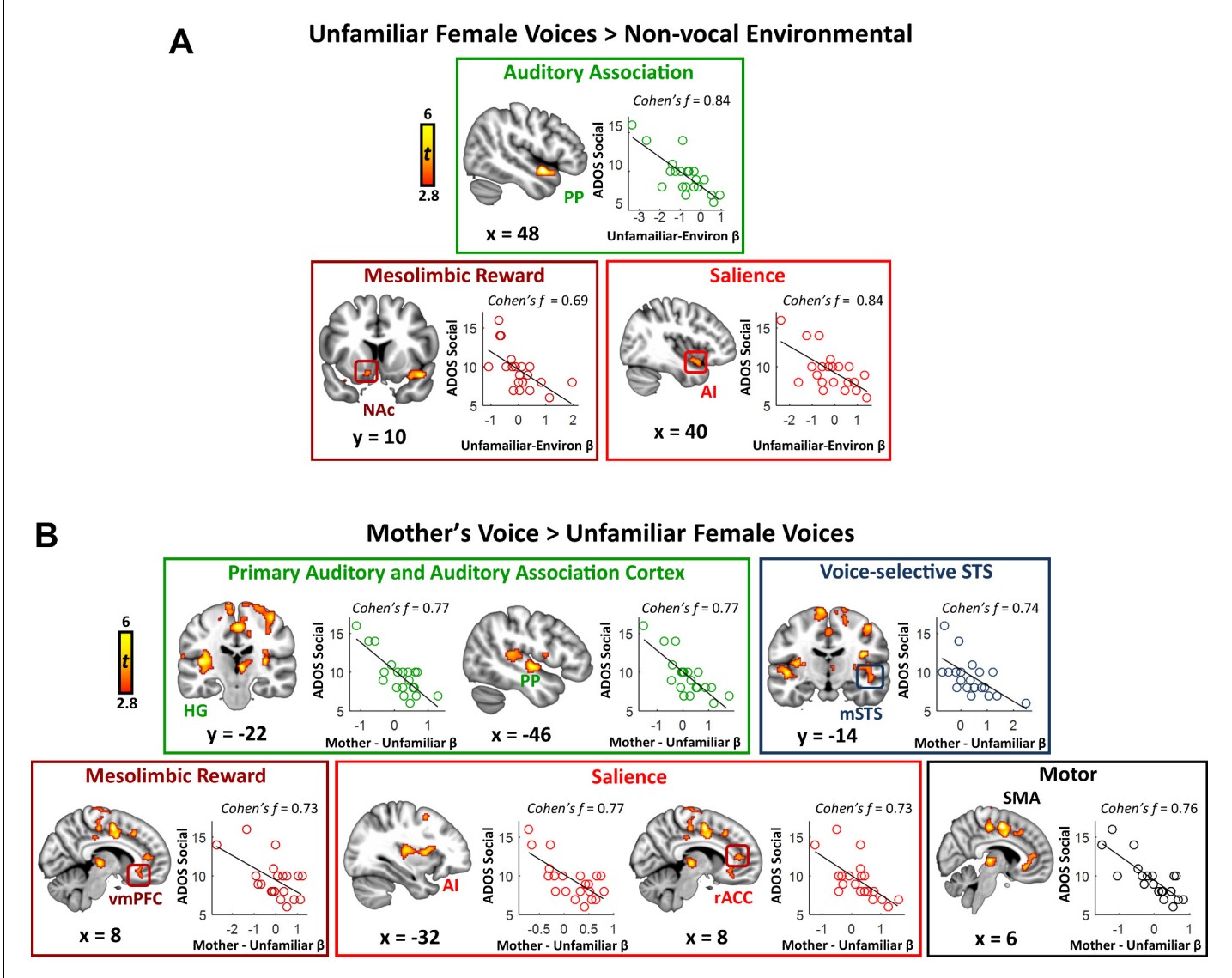

**Figure 3.** Activity in response to vocal stimuli and social communication abilities in children with ASD. (**A**) In children with ASD, the whole-brain covariate map shows that social communication scores are correlated with activity strength during unfamiliar female voice processing in auditory association cortex, the NAc of the reward system, and AI of the salience network. Scatterplots show the distributions and covariation of activity strength in response to unfamiliar female voices and standardized scores of social communication abilities in these children. Greater social communication abilities, reflected by smaller social communication scores, are associated with greater brain activity in these regions. (**B**) The whole-brain covariate map shows that social communication scores are correlated with activity strength during mother's voice processing in primary auditory and association cortex, voice-selective STS, vmPFC of the reward system, AI and rACC of the salience network, and SMA.
DOI: https://doi.org/10.7554/eLife.39906.004

## Connectivity patterns predict group membership

Functional connectivity was examined using a generalized psychophysiological interaction (gPPI) model within an extended voice processing brain network defined *a priori* from intrinsic connectivity results described in a previous study in children with ASD (*Abrams et al., 2013a*) (*Figure 4A*). This approach allows us to systematically build upon our previous findings while preempting task and sample-related biases in region-of-interest (ROI) selection. This extended voice processing network included ROIs in voice-selective STS, structures of the reward and salience networks, amygdala, hippocampus, and fusiform cortex (see *Appendix 1—table 3* for details of this network). There were no univariate group differences in individual links during either unfamiliar voice (*Figure 4B*) or mother's

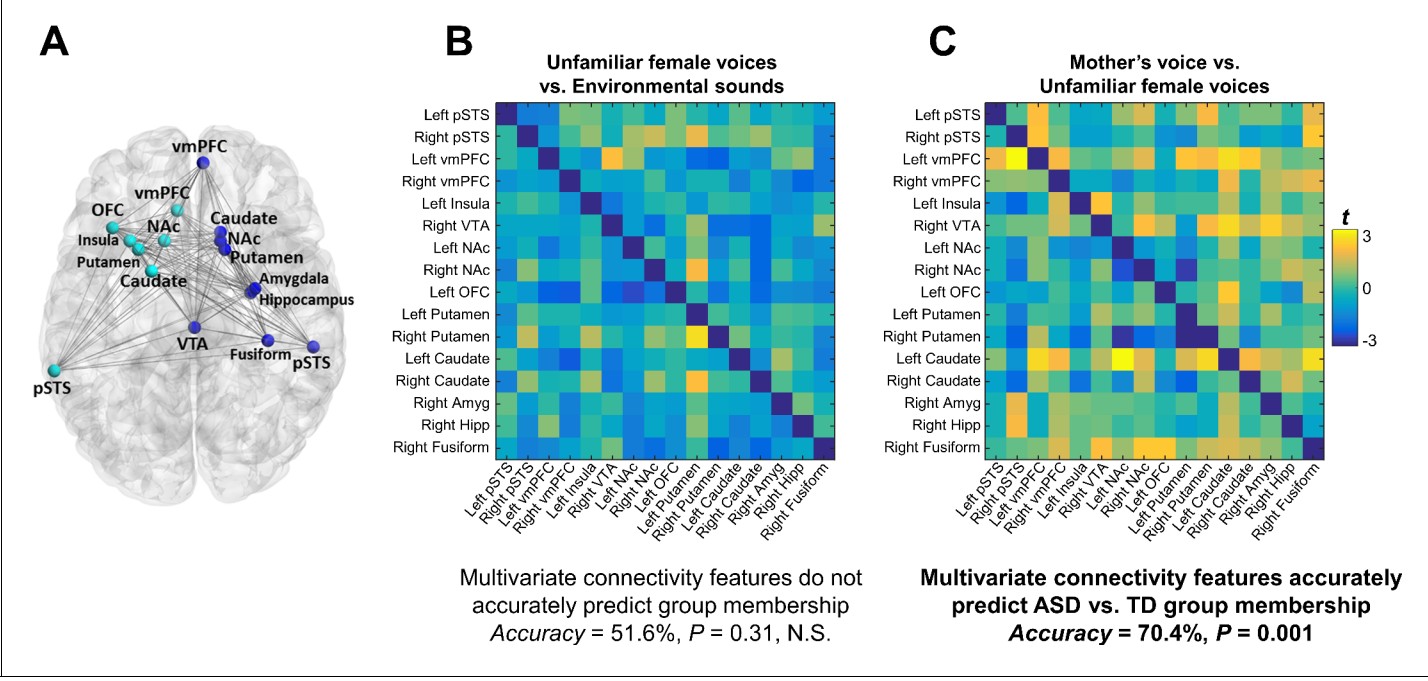

**Figure 4.** Functional connectivity in the extended voice-selective network and TD vs. ASD group membership. (**A**) The brain network used in connectivity analyses, which includes voice-selective, reward, salience, affective, and face-processing regions, was defined *a priori* from intrinsic connectivity results described in a previous study of children with ASD (*Abrams et al., 2013a*). (**B-C**) Group difference connectivity matrices shows differences in connectivity between TD children and children with ASD for all node combinations during (**B**) unfamiliar female voice processing and (**C**) mother's voice processing. Results from multivariate connectivity analysis show that connectivity patterns during mother's voice processing can accurately predict TD vs. ASD group membership; however, connectivity patterns during unfamiliar female voice processing are unable to accurately predict group membership.

DOI: https://doi.org/10.7554/eLife.39906.005

voice processing (*Figure 4C*) after correcting for multiple comparisons (FDR, *q* < 0.05). Support vector classification (SVC) results showed that multivariate connectivity patterns during unfamiliar voice processing were unable to predict group membership above chance (SVC Accuracy = 51.6%, p = 0.31); however, multivariate connectivity patterns during mother's voice processing accurately predicted TD vs. ASD group membership (SVC Accuracy = 70.4%, p = 0.001). We performed a confirmatory analysis using a different logistic regression classifier (GLMnet, generalized linear model via penalized maximum likelihood) and results were similar to the SVC results (unfamiliar voice processing: 54.8%, p = 0.78 (not significant); mother's voice: 80.9%, p = 0.010). These SVC results held even after accounting for group differences in mother's voice identification accuracy (see Appendix, *fMRI activation and connectivity profiles in children with ASD are not related to mother's voice identification accuracy*). Results show that patterns of brain connectivity during biologically-salient voice processing, but not unfamiliar voice processing, can distinguish children with ASD from TD children.

## Connectivity patterns predict social communication abilities

We next examined the relation between connectivity beta weights in each cell of the connectivity matrix and social communication scores in children with ASD. There were no significant univariate correlations between the strength of brain connectivity during either unfamiliar (*Figure 5B*) or mother's voice processing (*Figure 5C*) and social communication abilities. We then performed support vector regression (SVR) to examine whether multivariate patterns of connectivity during voice processing accurately predict social communication abilities in these children. Given that brain activation results showed that both unfamiliar (*Figure 3A*) and mother's voice processing (*Figure 3B*) explained variance in social communication abilities, we used a combination of connectivity features from both vocal conditions for this analysis. SVR results showed that multivariate connectivity patterns during unfamiliar and mother's voice processing accurately predict social communication

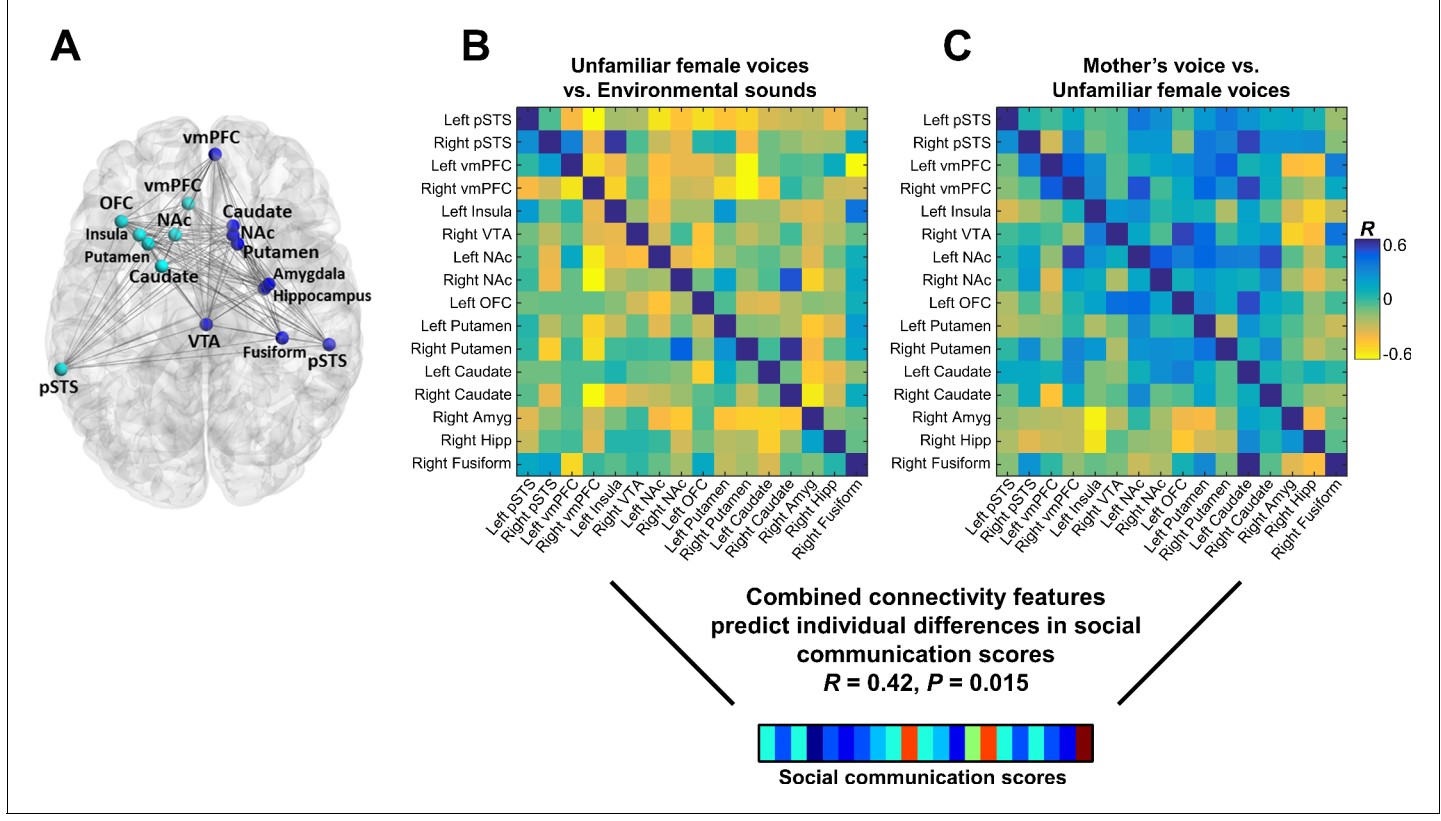

**Figure 5.** Functional connectivity in the extended voice-selective network and social communication abilities in children with ASD. (A) The brain network used in connectivity analyses, which includes voice-selective, reward, salience, affective, and face-processing regions, was defined *a priori* from intrinsic connectivity results described in a previous study of children with ASD (***Abrams et al., 2013a***). (B-C) Correlation matrices show Pearson's correlations between social communication scores and connectivity for each pairwise node combination in response to (B) unfamiliar female voice processing and (C) mother's voice processing in children with ASD. Results from multivariate connectivity analysis show that using a combination of connectivity features from both unfamiliar female and mother's voice processing can accurately predict social communication scores in children with ASD.

DOI: https://doi.org/10.7554/eLife.39906.006

scores in children with ASD ($R$ = 0.42, p = 0.015). We performed a confirmatory analysis using GLMnet and results were similar to the SVR results (social communication prediction: $R$ = 0.76, p < 0.001). Furthermore, when children with below chance accuracy on the mother's voice identification accuracy were removed from the analysis, this result held and connectivity patterns were still predictive of social communication scores (see Appendix, *fMRI activation and connectivity profiles in children with ASD are not related to mother's voice identification accuracy*).

## Discussion

It is unknown why children with ASD often 'tune out' from the voices of social partners in their environment (***Kanner, 1968***), including personal relations such as family members and caregivers (***Klin, 1991***). Here, we identify a striking relationship between individuals' social communication abilities and the strength of activation in reward and salience processing brain regions, notably NAc and AI, during human voice processing in children with ASD. Multivariate connectivity patterns within an extended voice processing network distinguished children with ASD from their TD peers and predicted social communication abilities in children with ASD. These findings suggest that dysfunction of the brain's reward system provides a stable brain signature of ASD that contributes to aberrant processing of salient vocal information (***Abrams et al., 2013a***).

## Regional and network features associated with voice processing predict individual differences in social function in children with ASD

Individuals with ASD present with a complex behavioral profile, which includes an array of sensory (*Marco et al., 2011*), cognitive (*Mundy and Newell, 2007*), and affective processing differences (*Harms et al., 2010*) compared to TD individuals. Consensus on the specific factors that most contribute to pronounced social communication difficulties in this population has remained elusive. Our findings showed that both regional and network features associated with voice processing, encompassing voice-selective cortex in the STS and extended voice-processing network that includes auditory, reward, and salience regions, predicted social function in children with ASD. The diversity of this network reflects the complexities of social communication itself, which involves the ability to integrate sensory, affective, mnemonic, and reward information. Importantly, our results unify several important characteristics of ASD in the extant literature, including regional functional aberrancies within specific brain systems and their association with social abilities (*Gervais et al., 2004*; *Kleinhans et al., 2008*; *Scott-Van Zeeland et al., 2010*; *Richey et al., 2014*; *Lombardo et al., 2015*), network level dysfunction (*Di Martino et al., 2011*; *Uddin et al., 2013b*; *von dem Hagen et al., 2013*; *Abrams et al., 2013a*), and heterogeneity of social communication abilities (*Lord et al., 2012*; *Lord et al., 1994*). We suggest that social communication function – human's ability to interact with and relate to others – is a unifying factor for explaining regional activation profiles and large-scale connectivity patterns linking key elements of the social brain.

## A voice-related brain network approach for understanding social information processing in autism

Brain network analyses represent an important approach for understanding brain function in autism (*Di Martino et al., 2011*; *Uddin et al., 2013b*; *von dem Hagen et al., 2013*; *Abrams et al., 2013a*), and psychopathology more broadly (*Menon, 2011*). These methods, which are typically applied to resting-state brain imaging data, have yielded considerable knowledge regarding network connectivity patterns in ASD and their links to behavior (*Abrams et al., 2013a*). A central assumption of this approach is that aberrant task-evoked circuit function is associated with clinical symptoms and behavior; however, empirical studies examining these associations have been lacking from the ASD literature. Our study addresses this gap by probing task-evoked function within a network defined *a priori* from a previous study of intrinsic connectivity of voice-selective networks in an independent group of children with ASD. We show that voice-related network function during the processing of a clinically and biologically meaningful social stimulus predicts both ASD group membership as well as social communication abilities in these children. Our findings bridge a critical gap between the integrity of the intrinsic architecture of the voice-processing network in children with ASD and network signatures of aberrant social information processing in these individuals.

## Biologically-salient vocal stimuli for investigating the social brain in autism spectrum disorders

Our results demonstrate that brief samples of a biologically salient voice, mother's voice, elicit a distinct neural signature in children with ASD. Our findings have important implications for the development of social skills in children with ASD. Specifically, typically developing children prefer biologically salient voices such as a mother's voice which provide critical cues for social (*Adams and Passman, 1979*) and language learning (*Liu et al., 2003*). In contrast, both anecdotal (*Kanner, 1968*) and experimental accounts (*Klin, 1991*) indicate that children with ASD do not show a preference for these sounds. We suggest that aberrant function within the extended voice processing network may underlie insensitivity to biologically salient voices in children with ASD, which may subsequently affect key developmental processes associated with social and pragmatic language learning.

## The social motivation theory and reward circuitry in children with ASD

The social motivation theory of ASD provides an important framework for considering pervasive social deficits in affected individuals (*Dawson et al., 2002*; *Chevallier et al., 2012*). The theory posits that social skills emerge in young children from an initial attraction to social cues in their environment. For example, TD infants are highly attentive to speech despite having no understanding of words' meanings, and this early attraction to vocal cues may be a critical step in a developmental

process that includes speech sound discrimination, mimicry, and, ultimately, language learning and verbal communication (*Kuhl et al., 2005b*). In contrast, children with ASD often do not engage with the speech in their environment (*Kanner, 1968*), and a central hypothesis of the social motivation theory is that weak reward attribution to vocal sounds during early childhood disrupts important developmental processes supporting social communication.

Our findings provide support for the social motivation theory by showing a link between social communication abilities in children with ASD and the strength of activity in reward and salience detection systems in response to unfamiliar and mother's voice. Specifically, children with ASD who have the most severe social communication deficits have the weakest responses in reward and salience detection brain regions to both of these vocal sources. Moreover, network connectivity of an extended voice-selective network, which includes nodes of the salience and reward networks, distinguished ASD and TD children and predicted social communication abilities in children with ASD. These results are the first to show that aberrant function of reward circuitry during voice processing is a distinguishing feature of childhood autism, and may limit the ability of children with ASD to experience vocal sounds as rewarding or salient. Our findings add to a growing literature suggesting that functional connectivity between voice-selective STS and reward and salience processing regions is an important predictor of social skill development in children (*Abrams et al., 2016*; *Abrams et al., 2013a*).

Our results highlighting the role of reward and salience in the context of voice processing have implications for clinical treatment of social communication deficits in children with ASD. An important direction for treatment of children with ASD involves the use of teaching strategies (*Dawson et al., 2010*; *Koegel and Koegel, 2006*) that focus on motivating children to engage in verbal interactions to improve social communication skills (*Koegel et al., 2005*; *Mundy and Stella, 2000*). Findings suggest that clinical efforts to increase the reward value of vocal interactions in children with ASD may be key to remediating social communication deficits in these individuals. Furthermore, neural activity and connectivity measures may represent a quantitative metric for assessing response to clinical treatments focused on verbal interactions.

## Limitations

There are limitations to the current work that warrant consideration. First, the sample size is relatively modest compared to recent task-based brain imaging studies of neurotypical adult populations and resting-state fMRI or structural MRI studies in individuals with ASD, however these types of studies do not face the same data collection challenges as task-based studies in clinical pediatric populations (*Yerys et al., 2009*). Importantly, resting-state and structural imaging studies are unable to address specific questions related to social information processing in ASD, such as biologically salient voice processing, which are critical for understanding the brain bases of social dysfunction in affected children. Indeed, our sample size is larger than, or comparable to, the majority of task-fMRI studies in children with ASD published since 2017, and have more stringent individual-level sampling compared to these studies. This is an important consideration given that the replicability of task fMRI data is not solely contingent on a large sample size but also depends on the amount of individual-level sampling. A recent report examining this question showed that modest sample sizes, comparable to those described in our submitted manuscript, yield highly replicable results with only four runs of task data with a similar number of trials per run as our study (*Nee, 2018*). In comparison, we required that each child participant had at least seven functional imaging runs of our event-related fMRI task that met our strict head movement criteria. A final limitation of this work is that, consistent with the vast majority of brain imaging studies in children with ASD, we were unable to include lower functioning children with ASD since the scanner environment is ill-suited for these children (*Yerys et al., 2009*). Further studies with larger samples are needed both to capture the full range of heterogeneity of ASD and to ensure the broader generalizability of the findings reported here.

## Conclusion

We identified neural features underlying voice processing impairments in children with ASD, which are thought to contribute to pervasive social communication difficulties in affected individuals. Results show that activity profiles and network connectivity patterns within voice-selective and

reward regions, measured during unfamiliar and mother's voice processing, distinguish children with ASD from TD peers and predict their social communication abilities. These findings are consistent with the social motivation theory of ASD by linking human voice processing to dysfunction in the brain's reward centers, and have implications for the treatment of social communication deficits in children with ASD. For example, parent training has emerged as a powerful and cost-effective approach for increasing treatment intensity (*National Research Council, 2001*): treatment delivery in the child's natural environment promotes functional communication (*Delprato, 2001*), generalization (*Stokes and Baer, 1977*), and maintenance of skills over time (*Sheinkopf and Siegel, 1998*; *Moes and Frea, 2002*). Findings from the current study, which demonstrate a link between social communication function and neural processing of mother's voice, support the importance of parent training by suggesting that a child's ability to focus on, and direct neural resources to, these critical communication partners may be a key to improving social function in affected children.

## Materials and methods

### Participants

The Stanford University Institutional Review Board approved the study protocol. Parental consent and the child's assent were obtained for all evaluation procedures, and children were paid for their participation in the study.

A total of 57 children were recruited from around the San Francisco Bay Area for this study. All children were required to be right-handed and have a full-scale IQ > 80, as measured by the Wechsler Abbreviated Scale of Intelligence (WASI) (*Wechsler, 1999*). 28 children met ASD criteria based on an algorithm (*Risi et al., 2006*) that combines information from both the module 3 of the ADOS-2 (47) and the ADI–Revised (*Lord et al., 1994*). Specifically, these children showed mild to more severe social communication deficits, particularly in the areas of social-emotional reciprocity and verbal and non-verbal communication, and repetitive and restricted behaviors and interests (*American Psychiatric Association, 2013*). Five children with ASD were excluded because of excessive movement in the fMRI scanner, one child was excluded because of a metal retainer interfering with their brain images, and one child was excluded because their biological mother was not available to do a voice recording. Importantly, children in the ASD sample are considered 'high-functioning' and had fluent language skills and above-average reading skills (*Table 1*). Nevertheless, these children are generally characterized as having communication impairments, especially in the area of reciprocal conversation.

TD children and had no history of neurological, psychiatric, or learning disorders, personal and family history (first degree) of developmental cognitive disorders and heritable neuropsychiatric disorders, evidence of significant difficulty during pregnancy, labor, delivery, or immediate neonatal period, or abnormal developmental milestones as determined by neurologic history and examination. Three TD children were excluded because of excessive movement in the fMRI scanner, one was excluded because of scores in the 'severe' range on standardized measures of social function, and four female TD children were excluded to provide a similar ratio of males to females relative to the ASD participants. The final TD and ASD groups that were included in the analysis consisted of 21 children in each group who were matched for full-scale IQ, age, sex, and head motion during the fMRI scan (*Table 1*). All participants are the biological offspring of the mothers whose voices were used in this study (i.e. none of our participants were adopted, and therefore none of the mother's voices are from an adoptive mother), and all participants were raised in homes that included their mothers. Participants' neuropsychological characteristics are provided in *Table 1*.

### Data acquisition parameters

All fMRI data were acquired at the Richard M. Lucas Center for Imaging at Stanford University. Functional images were acquired on a 3 T Signa scanner (General Electric) using a custom-built head coil. Participants were instructed to stay as still as possible during scanning, and head movement was further minimized by placing memory-foam pillows around the participant's head. A total of 29 axial slices (4.0 mm thickness, 0.5 mm skip) parallel to the anterior/posterior commissure line and covering the whole brain were imaged by using a T2*-weighted gradient-echo spiral in-out pulse sequence (*Glover and Law, 2001*) with the following parameters: repetition time = 3576 ms; echo time = 30

ms; flip angle = 80°; one interleave. The 3576 msec TR can be calculated as the sum of: (1) the stimulus duration of 956 msec; (2) a 300 ms silent interval buffering the beginning and end of each stimulus presentation (600 ms total of silent buffers) to avoid backward and forward masking effects; (3) the 2000 ms volume acquisition time; and (4) an additional 20 ms silent interval, which helped the stimulus computer maintain precise and accurate timing during stimulus presentation. The field of view was 20 cm, and the matrix size was 64 × 64, providing an in- plane spatial resolution of 3.125 mm. Reduction of blurring and signal loss arising from field inhomogeneities was accomplished by the use of an automated high-order shimming method before data acquisition.

## fMRI Task

Auditory stimuli were presented in 10 separate runs, each lasting 4 min. One run consisted of 56 trials of mother's voice, unfamiliar female voices, environmental sounds and catch trials, which were pseudo-randomly ordered within each run. Stimulus presentation order was the same for each subject. Each stimulus lasted 956 msec in duration. Prior to each run, child participants were instructed to play the 'kitty cat game' during the fMRI scan. While laying down in the scanner, children were first shown a brief video of a cat and were told that the goal of the cat game was to listen to a variety of sounds, including 'voices that may be familiar,' and to push a button on a button box only when they heard kitty cat meows (catch trials). The function of the 'catch trials' was to keep the children alert and engaged during stimulus presentation. During each run, four or five exemplars of each stimulus type (i.e. nonsense words samples of mother's and unfamiliar female voices, environmental sounds), as well as three catch trials, were presented. At the end of each run, the children were shown another engaging video of a cat. Although the button box failed to register responses during data collection in four children with ASD and nine TD children, data analysis of the catch trails for 17 children with ASD and 12 TD children showed similar catch trial accuracies between TD (accuracy = 91%) and ASD groups (accuracy = 89%; two-sample $t$-test results: $t(2) = 0.35$, p = 0.73). Across the ten runs, a total of 48 exemplars of each stimulus condition were presented to each subject (i.e. 144 total exemplars produced by each of the three vocal sources, including the child's mother, unfamiliar female voice #1, and unfamiliar female voice #2). Vocal stimuli were presented to participants in the scanner using Eprime V1.0 (Psychological Software Tools, 2002). Participants wore custom-built headphones designed to reduce the background scanner noise to ~70 dBA (*Abrams et al., 2011*; *Abrams et al., 2013b*). Headphone sound levels were calibrated prior to each data collection session, and all stimuli were presented at a sound level of 75 dBA. Participants were scanned using an event-related design. Auditory stimuli were presented during silent intervals between volume acquisitions to eliminate the effects of scanner noise on auditory discrimination. One stimulus was presented every 3576 ms, and the silent period duration was not jittered. The total silent period between stimulus presentations was 2620 ms, and consisted of a 300 ms silent period, 2000 ms for a volume acquisition, another 300 ms of silence, and a 20 ms silent interval that helped the stimulus computer maintain precise and accurate timing during stimulus presentation.

## Functional MRI preprocessing

fMRI data collected in each of the 10 functional runs were subject to the following preprocessing procedures. The first five volumes were not analyzed to allow for signal equilibration. A linear shim correction was applied separately for each slice during reconstruction by using a magnetic field map acquired automatically by the pulse sequence at the beginning of the scan. Translational movement in millimeters (x, y, z) was calculated based on the SPM8 parameters for motion correction of the functional images in each subject. To correct for deviant volumes resulting from spikes in movement, we used a de-spiking procedure. Volumes with movement exceeding 0.5 voxels (1.562 mm) or spikes in global signal exceeding 5% were interpolated using adjacent scans. The majority of volumes repaired occurred in isolation. After the interpolation procedure, images were spatially normalized to standard Montreal Neurological Institute (MNI) space, resampled to 2 mm isotropic voxels, and smoothed with a 6 mm full-width at half maximum Gaussian kernel.

## Movement criteria for inclusion in fMRI analysis

For inclusion in the fMRI analysis, we required that each functional run had a maximum scan-to-scan movement of < 6 mm and no more than 15% of volumes were corrected in the de-spiking

procedure. Moreover, we required that all individual subject data included in the analysis consisted of at least seven functional runs that met our criteria for scan-to-scan movement and percentage of volumes corrected; subjects who had fewer than seven functional runs that met our movement criteria were not included in the data analysis. All 42 participants included in the analysis had at least seven functional runs that met our movement criteria, and the total number of runs included for TD and ASD groups were similar (TD = 192 runs; ASD = 188 runs).

## Voxel-wise analysis of fMRI activation

The goal of this analysis was to identify brain regions that showed differential activity levels in response to mother's voice, unfamiliar voices, and environmental sounds. Brain activation related to each vocal task condition was first modeled at the individual subject level using boxcar functions with a canonical hemodynamic response function and a temporal derivative to account for voxel-wise latency differences in hemodynamic response. Environmental sounds were not modeled to avoid collinearity, and this stimulus served as the baseline condition. Low-frequency drifts at each voxel were removed using a high-pass filter (0.5 cycles/min) and serial correlations were accounted for by modeling the fMRI time series as a first-degree autoregressive process (*Friston et al., 1997*). We performed whole-brain ANOVAs to separately investigate unfamiliar and mother's voice processing: (1) the unfamiliar voice analysis used the factors group (TD and ASD) and auditory condition (unfamiliar voices and environmental sounds) and (2) the mother's voice analysis used the factors group (TD and ASD) and voice condition (mother's voice and unfamiliar voices). These ANOVAs were designed to test specific hypotheses described in the Introduction. Group-level activation was determined using individual subject contrast images and a second-level analysis of variance. The main contrasts of interest were [mother's voice – unfamiliar female voices] and [unfamiliar female voices – environmental sounds]. Significant clusters of activation were determined using a voxel-wise statistical height threshold of p < 0.005, with family-wise error corrections for multiple spatial comparisons (p < 0.05; 67 voxels) determined using Monte Carlo simulations (*Forman et al., 1995*; *Ward, 2000*) using a custom Matlab script (see Source Code). To examine GLM results in the inferior colliculus and NAc, small subcortical brain structures, we used a small volume correction at p<0.05 with a voxel-wise statistical height threshold of p < 0.005. To determine the robustness of our findings, group comparisons were also performed using more stringent height and extent thresholds (*Appendix 1—tables 4–5*). To provide estimates of effect sizes within specific regions displayed in *Figure 2*, t-scores from the whole-brain TD vs. ASD group GLM analysis were averaged within each significant cluster. Effect sizes were then computed as Cohen's d according to *Equation 1* below, where t is the mean t-score within a cluster and N is the sample size:

$$Cohen's\ d = \frac{t}{sqrt\left(\frac{N}{2}\right)} \tag{1}$$

To define specific cortical regions, we used the Harvard–Oxford probabilistic structural atlas (*Smith et al., 2004*) with a probability threshold of 25%.

## Brain-behavior analysis

Regression analysis was used to examine the relationship between brain responses to unfamiliar and mother's voice and social communication abilities in children with ASD. Social communication function was assessed using the Social Affect subscore of the ADOS-2 (47). Brain-behavior relationships were examined using analysis of activation levels. A whole-brain, voxel-wise regression analysis was performed in which the relation between fMRI activity and social communication scores was examined using images contrasting [unfamiliar female voices > environmental sounds] and [mother's vs. unfamiliar female voices]. Significant clusters were determined using a voxel-wise statistical height threshold of p < 0.005, with family-wise error corrections for multiple spatial comparisons (p < 0.05; 67 voxels) determined using Monte Carlo simulations (*Forman et al., 1995*; *Ward, 2000*). To determine the robustness of our findings, brain-behavior relations were also examined using more stringent height and extent thresholds (*Appendix 1—tables 6–7*). To provide estimates of effect sizes within regions displayed in *Figure 3*, t-scores from the whole-brain ASD Social Communication covariate analysis were averaged within each cluster identified in the GLM analysis. Effect sizes were

then computed as Cohen's *f* according to *Equation 2* below, where *t* is the mean *t*-score within a cluster and *N* is the sample size:

$$Cohen\, f = \frac{t}{sqrt(N)} \qquad (2)$$

## Brain activity levels and prediction of social function

To examine the robustness and reliability of brain activity levels for predicting social communication scores, we used support vector regression (SVR) to perform a confirmatory cross-validation analysis that employs a machine-learning approach with balanced fourfold cross-validation (CV) combined with linear regression (*Cohen et al., 2010*). In this analysis, we extracted individual subject activation beta values taken from the [unfamiliar female voices > environmental sounds] and [mother's voice > unfamiliar female voices] GLM contrasts. For the [unfamiliar female voices > environmental sounds] GLM contrast, GLM betas were extracted from right-hemisphere PP and AI as well as left-hemisphere NAc. For the [mother's voice > unfamiliar female voices] GLM contrast, GLM betas were extracted from left-hemisphere HG, PP, and AI as well as right-hemisphere mSTS, vmPFC, rACC, and SMA. These values were entered as independent variables in a linear regression analysis with ADOS-2 Social Affect subscores as the dependent variable. r $_{(predicted,\ observed)}$, a measure of how well the independent variable predicts the dependent variable, was first estimated using a balanced fourfold CV procedure. Data were divided into four folds so that the distributions of dependent and independent variables were balanced across folds. Data were randomly assigned to four folds and the independent and dependent variables tested in one-way ANOVAs, repeating as necessary until both ANOVAs were insignificant in order to guarantee balance across the folds. A linear regression model was built using three folds leaving out the fourth, and this model was then used to predict the data in the left-out fold. This procedure was repeated four times to compute a final r$_{(predicted, observed)}$ representing the correlation between the data predicted by the regression model and the observed data. Finally, the statistical significance of the model was assessed using a nonparametric testing approach. The empirical null distribution of r $_{(predicted,\ observed)}$ was estimated by generating 1000 surrogate datasets under the null hypothesis that there was no association between changes in ADOS social communication subscore and brain activity levels.

## Functional connectivity analysis

We examined functional connectivity between ROIs using the generalized psychophysiological interaction (gPPI) model (*McLaren et al., 2012*), with the goal of identifying connectivity between ROIs in response to each task condition as well differences between task conditions (mother's voice, other voice, environmental sounds). We used the SPM gPPI toolbox for this analysis. gPPI is more sensitive than standard PPI to task context-dependent differences in connectivity (*McLaren et al., 2012*). Unlike dynamical causal modeling (DCM), gPPI does not use a temporal precedence model (x (t + 1)~x(t)) and therefore makes no claims of causality. The gPPI model is summarized in *Equation 3* below:

$$ROI_{target} \sim conv\big(deconv(ROI_{seed}) * task_{waveform}\big) + ROI_{seed} + constant \qquad (3)$$

Briefly, in each participant, the regional timeseries from a seed ROI was deconvolved to uncover quasi-neuronal activity and then multiplied with the task design waveform for each task condition to form condition-specific gPPI interaction terms. These interaction terms are then convolved with the hemodynamic response function (HRF) to form gPPI regressors for each task condition. The final step is a standard general linear model predicting target ROI response after regressing out any direct effects of the activity in the seed ROI. In the equation above, $ROI_{target}$ and $ROI_{seed}$ are the time series in the two brain regions, and $task_{waveform}$ contains three columns corresponding to each task condition. The goal of this analysis was to examine connectivity patterns within an extended voice-selective network identified in a previous study of children with ASD (*Abrams et al., 2013a*). This study showed weak intrinsic connectivity between bilateral voice-selective STS and regions implicated in reward, salience, memory, and affective processing. The rationale for the use of an *a priori* network is it is an established method of network identification that preempts task and sample-related biases in region-of-interest (ROI) selection. This approach therefore allows for a more generalizable set of results compared to a network defined based on nodes identified using the current

sample of children and task conditions. The network used in all connectivity analyses consisted of 16 regions. All cortical ROIs were constructed as 5 mm spheres centered on the coordinates listed in *Appendix 1—table 3*, while subcortical ROIs were constructed as 2 mm spheres.

## Functional connectivity, group classification, and prediction of social function

Support vector classification (SVC) and regression (SVR) were used to examine whether patterns of connectivity within the extended voice processing network could predict TD vs. ASD group membership and social communication abilities in children with ASD, respectively. First, to examine TD vs. ASD group membership, a linear support vector machine algorithm (C = 1) from the open-source library LIBSVM (http://www.csie.ntu.edu.tw/~cjlin/libsvm/) was used to build classifiers to distinguish children with ASD from TD children during unfamiliar voice processing. Individual subject connectivity matrices (16 × 16 ROIs) taken from the [unfamiliar female voices > environmental sounds] gPPI contrast were used as features to train classifiers in each dataset. Classifier performance was evaluated using a four-fold cross-validation procedure. Specifically, a dataset was randomly partitioned into four folds. Three folds of data (training set) were used to train a classifier, which was then applied to the remaining fold (test set) to predict whether each sample in the test set should be classified as ASD or TD. This procedure was repeated four times with each of the four folds used exactly once as a test set. The average classification accuracy across the four folds (cross-validation accuracy) was used to evaluate the classifier's performance. To further account for variation due to random data partition, we repeated the same cross-validation procedure 100 times with different random data partitions. Finally, the mean cross-validation accuracies from 100 iterations was reported, and its statistical significance was evaluated using permutation testing (1000 times) by randomly permuting subjects' labels and repeating the same above procedures. The same SVC methods were used to examine whether connectivity features during mothers voice processing could accurately predict TD vs. ASD group membership, however in this analysis individual subject connectivity matrices (16 × 16 ROIs) taken from the [mother's voice > unfamiliar female voices] gPPI contrast were used as features to train the classifier.

Finally, SVR was used to examine whether connectivity patterns during unfamiliar female and mother's voice processing could predict social communication scores in children with ASD. SVR methods are the same as those described in *Brain Activity Levels and Prediction of Social Function*; however, features in this analysis include multivariate connectivity patterns across the extended voice-selective network (16 ROIs). Given that brain activation results showed that both unfamiliar (*Figure 3A*) and mother's voice processing (*Figure 3B*) explained variance in social communication abilities, we used a combination of connectivity features from both vocal conditions for this analysis. Specifically, connectivity features from both the [unfamiliar female voices > environmental sounds] and [mother's voice > unfamiliar female voices] gPPI contrasts were entered as independent variables in a linear regression analysis with ADOS-2 Social Affect subscores as the dependent variable.

As a confirmatory analysis, and to examine the robustness of SVC and SVR results, we used GLMnet (http://www-stat.stanford.edu/~tibs/glmnet-matlab), a logistic regression classifier that includes regularization and exploits sparsity in the input matrix, on the same 16 × 16 connectivity matrices described for the SVC and SVR analyses above.

## Stimulus design considerations

Previous studies investigating the processing (*DeCasper and Fifer, 1980*; *Adams and Passman, 1979*) and neural bases (*Imafuku et al., 2014*; *Purhonen et al., 2004*) of mother's voice processing have used a design in which one mother's voice serves as a control voice for another participant. However, due to an important practical limitation, the current study used a design in which all participants heard the same two control voices. While we make every effort to recruit children from a variety of communities in the San Francisco Bay Area, some level of recruitment occurs through contact with specific schools, and in other instances our participants refer their friends to our lab for inclusion in our studies. In these cases, it is a reasonable possibility that our participants may have known other mothers involved in the study, and therefore may be familiar with these mothers' voices, which would limit the control we were seeking in our control voices. Importantly, HIPPA guidelines are explicit that participant information is confidential, and therefore there would be no way to probe

whether a child knows any of the other families involved in the study. Given this practical consideration, we concluded that it would be best to use the same two control voices, which we knew were unfamiliar to the participants, for all participants' data collection.

## Stimulus recording

Recordings of each mother were made individually while their child was undergoing neuropsychological testing. Mother's voice stimuli and control voices were recorded in a quiet conference room using a Shure PG27-USB condenser microphone connected to a MacBook Air laptop. The audio signal was digitized at a sampling rate of 44.1 kHz and A/D converted with 16-bit resolution. Mothers were positioned in the conference room to avoid early sound wave reflections from contaminating the recordings. To provide a natural speech context for the recording of each nonsense word, mothers were instructed to repeat three sentences, each of which contained one of the nonsense words, during the recording. The first word of each of these sentence was their child's name, which was followed by the words 'that is a,' followed by one of the three nonsense words. A hypothetical example of a sentence spoken by a mother for the recording was 'Johnny, that is a keebudieshawlt.' Prior to beginning the recording, mothers were instructed on how to produce these nonsense words by repeating them to the experimenter until the mothers had reached proficiency. Importantly, mothers were instructed to say these sentences using the tone of voice they would use when speaking with their child during an engaging and enjoyable shared learning experience (e.g. if their child asked them to identify an item at a museum). The vocal recording session resulted in digitized recordings of the mothers repeating each of the three sentences approximately 30 times to ensure multiple high-quality samples of each nonsense word for each mother.

## Stimulus post-processing

The goal of stimulus post-processing was to isolate the three nonsense words from the sentences that each mother spoke during the recording session and normalize them for duration and RMS amplitude for inclusion in the fMRI stimulus presentation protocol and the mother's voice identification task. First, a digital sound editor (Audacity: http://audacity.sourceforge.net/) was used to isolate each utterance of the three nonsense words from the sentences spoken by each mother. The three best versions of each nonsense word were then selected based on the audio and vocal quality of the utterances (i.e. eliminating versions that were mispronounced, included vocal creak, or were otherwise not ideal exemplars of the nonsense words). These nine nonsense words were then normalized for duration to 956 ms, the mean duration of the nonsense words produced by the unfamiliar female voices, using Praat software similar to previous studies (*Abrams et al., 2016*; *Abrams et al., 2008*). A 10 msec linear fade (ramp and damp) was then performed on each stimulus to prevent click-like sounds at the beginning and end of the stimuli, and then stimuli were equated for RMS amplitude. These final stimuli were then evaluated for audibility and clarity to ensure that post-processing manipulations had not introduced any artifacts into the samples. The same process was performed on the control voices and environmental sounds to ensure that all stimuli presented in the fMRI experiment were the same duration and RMS amplitude.

## Post-scan mother's voice identification task

All participants who participated in the fMRI experiment completed an auditory behavioral test following the fMRI scan. The goal of the Mother's Voice Identification Task was to determine if the participants could reliably discriminate their mother's voice from unfamiliar female voices. Participants were seated in a quiet room in front of a laptop computer, and headphones were placed over their ears. In each trial, participants were presented with a recording of a multisyllabic nonsense word spoken by either the participant's mother or a control mother, and the task was to indicate whether or not their mother spoke the word. The multisyllabic nonsense words used in the behavioral task were the exact same samples used in the fMRI task. Each participant was presented with 54 randomly ordered nonsense words: 18 produced by the subject's mother and the remaining 36 produced by unfamiliar female voices.

## Signal level analysis

Group mean activation differences for key brain regions identified in the whole-brain univariate analysis were calculated to examine the basis for TD > ASD group differences for both [unfamiliar female voices > environmental sounds] (*Figure 2A*) and [mother's voice > unfamiliar female voices] contrasts (*Figure 2B*). The reason for this analysis is that stimulus differences can result from a number of different factors. For example, both mother's voice and unfamiliar female voices could elicit reduced activity relative to baseline and significant stimulus differences could be driven by greater negative activation in response to unfamiliar female voices. Significant stimulus differences were inherent to this ROI analysis as they are based on results from the whole-brain GLM analysis (*Vul et al., 2009*); however, results provide important information regarding the magnitude and sign of results in response to both stimulus conditions. Baseline for this analysis was calculated as the brain response to environmental sounds. The coordinates for the ROIs used in the signal level analysis were based on peaks in TD > ASD group maps for the [unfamiliar female voices > environmental sounds] and [mother's voice > unfamiliar female voices] contrasts. Cortical ROIs were defined as 5 mm spheres, and subcortical ROIs were 2 mm spheres, centered at the peaks in the TD > ASD group maps for the [unfamiliar female voices > environmental sounds] or [mother's voice >unfamiliar female voices] contrasts. Signal level was calculated by extracting the β-value from individual subjects' contrast maps for the [unfamiliar female voices > environmental sounds] and [mother's voice >environmental sounds] comparisons. The mean β-value within each ROI was computed for both contrasts in all subjects. The group mean β and its standard error for each ROI are plotted in *Appendix 1—figure 3*.

## Acknowledgements

This work was supported by NIH Grants K01 MH102428 (to DAA), and DC011095 and MH084164 (to VM), a NARSAD Young Investigator Grant from the Brain and Behavior Research Foundation (to DAA), Stanford Child Health Research Institute and the Stanford NIH-NCATS-CTSA (to DAA; UL1 TR001085), the Singer Foundation, and the Simons Foundation/SFARI (308939, VM). We thank all the children and their parents who participated in our study, E Adair and the staff at the Stanford Lucas Center for Imaging for assistance with data collection, S Karraker for assistance with data processing, H Abrams and C Anderson for help with stimulus production, and C Feinstein for helpful discussions.

## Additional information

### Funding

| Funder | Grant reference number | Author |
| --- | --- | --- |
| National Institute of Mental Health | MH102428 | Daniel Arthur Abrams |
| Brain and Behavior Research Foundation | NARSAD Young Investigator Grant | Daniel Arthur Abrams |
| Stanford School of Medicine, Stanford Medicine, Stanford University | UL1TR001085 | Daniel Arthur Abrams |
| National Center for Advancing Translational Sciences | UL1TR001085 | Daniel Arthur Abrams |
| National Institute on Deafness and Other Communication Disorders | DC011095 | Vinod Menon |
| National Institute of Mental Health | MH084164 | Vinod Menon |
| Singer Family Foundation | | Vinod Menon |
| Simons Foundation | 308939 | Vinod Menon |

The funders had no role in study design, data collection and interpretation, or the decision to submit the work for publication.

## Author contributions

Daniel Arthur Abrams, Conceptualization, Formal analysis, Supervision, Funding acquisition, Investigation, Visualization, Methodology, Writing—original draft, Project administration, Writing—review and editing; Aarthi Padmanabhan, Supervision, Writing—original draft, Project administration, Writing—review and editing; Tianwen Chen, John Kochalka, Formal analysis; Paola Odriozola, Amanda E Baker, Data curation, Investigation, Project administration; Jennifer M Phillips, Investigation; Vinod Menon, Conceptualization, Resources, Supervision, Funding acquisition, Writing—original draft, Project administration, Writing—review and editing

## Author ORCIDs

Daniel Arthur Abrams http://orcid.org/0000-0002-1255-1200
Aarthi Padmanabhan http://orcid.org/0000-0002-3727-5468
Paola Odriozola http://orcid.org/0000-0003-1641-4139
Amanda E Baker http://orcid.org/0000-0002-0140-2162
Jennifer M Phillips http://orcid.org/0000-0002-6360-2346

## Ethics

Human subjects: The Stanford University Institutional Review Board approved the study protocol (Protocol # 11849). Parental written informed consent and consent to publish were obtained for all participants, and the child's assent was obtained for all evaluation procedures. Children were paid for their participation in the study. All procedures performed were in accordance with ethical standards set out by the Federal Policy for the Protection of Human Subjects (or 'Common Rule', U.S. Department of Health and Human Services Title 45 DFR 46).

## Decision letter and Author response

Decision letter https://doi.org/10.7554/eLife.39906.028
Author response https://doi.org/10.7554/eLife.39906.029

## Additional files

### Supplementary files

• Source code 1. Spatial Extent with Monte Carlo Simulations.
DOI: https://doi.org/10.7554/eLife.39906.008

• Transparent reporting form
DOI: https://doi.org/10.7554/eLife.39906.009

### Data availability

All fMRI activation maps reported in the manuscript will be made available at NeuroVault (https://neurovault.org/collections/4815/). Full single subject raw data will be made public on the NIH NDAR repository, as per NIH rules (procedure is ongoing).

The following dataset was generated:

| Author(s) | Year | Dataset title | Dataset URL | Database and Identifier |
|---|---|---|---|---|
| Daniel Arthur Abrams, Aarthi Padmanabhan | 2019 | fMRI activation maps reported in 'Impaired voice processing in reward and salience circuits predicts social communication in children with autism' | https://neurovault.org/collections/4815/ | NeuroVault, 4815 |

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

# Appendix 1

DOI: https://doi.org/10.7554/eLife.39906.010

## Acoustical analysis of mother's voice samples

We performed acoustical analyses of mother's voice and unfamiliar voice samples to characterize the physical attributes of the stimuli used for fMRI data collection. The goal of these analyses was to determine if differences between vocal samples collected from mothers of children with ASD and those collected from mothers of TD controls could potentially account for group differences in fMRI activity. Human voices are differentiated according to several acoustical features, including those reflecting the anatomy of the speaker's vocal tract, such as the pitch and harmonics of speech, and learned aspects of speech production, which include speech rhythm, rate, and emphasis (*Bricker and Pruzansky, 1976*; *Hecker, 1971*). Acoustical analysis showed that vocal samples collected from mothers of children with ASD were comparable to those collected from mothers of TD controls measured across multiple spectrotemporal acoustical features (p > 0.10 for all acoustical measures; *Figure 1B*). An additional goal of the fMRI data analysis was to examine individual differences in social communication abilities in children with ASD, and therefore the next analysis focused on whether acoustical features varied as a function of social communication abilities in children with ASD; there was no relationship between acoustical measures and social communication scores (p > 0.25 for all acoustical measures). Finally, acoustical analyses of the unfamiliar voice samples used in all fMRI sessions were qualitatively similar to vocal samples collected from the mothers of TD controls and children with ASD. Together, these results indicate that there are no systematic differences in the acoustical properties of vocal samples collected from participants' mothers that could potentially bias the fMRI analysis.

## Identification of mother's voice

To examine whether children who participated in the fMRI study could identify their mother's voice accurately in the brief vocal samples used in the fMRI experiment, participants performed a mother's voice identification task. All TD children identified their mother's voice with a high degree of accuracy (mean accuracy = 97.5%; *Figure 1C*), indicating that brief (< 1 s) pseudoword speech samples are sufficient for the consistent and accurate identification of mother's voice in these children. 16 of the 21 children in the ASD sample were also able to identify their mother's voice with a high degree of accuracy (mean accuracy = 98.2%), however the remaining five children with ASD performed below chance on this task. Group comparison revealed that TD children had greater mother's voice identification accuracy compared to children with ASD ($t(40)$ = 2.13, p=0.039).

An important question is whether the five children with ASD who performed below chance on the mother's voice identification task might show a distinct behavioral signature that may help explain why these children were unable to identify their mother's voice in our identification task. While these children did not present with hearing impairments as noted by parents or neuropsychological assessors, who had performed extensive neuropsychological testing on these children prior to the fMRI scan and mother's voice identification task, a plausible hypothesis is that the five children who were unable to identify their mother's voice in the task would show greater social communication deficits, or lower scores on measures of cognitive and language function, and/or reduced brain activation in response to unfamiliar or mother's voice stimuli. To test this hypothesis, we performed additional analyses to examine whether there are any identifying clinical or cognitive characteristics regarding these five children with low mother's voice identification accuracy.

We first examined differences in social communication scores and measures of cognitive and language abilities between children with ASD with low (N = 5) vs. high (N = 16) mother's voice identification accuracy. Examining the distribution of ADOS Social scores revealed that the five children with low mother's voice identification accuracy had a wide range of scores from 7 to 16 (please note that ADOS Social Affect is scored in a range between 0–20, with a

score of 0 indicating no social deficit, a score of 7 indicating a more mild social communication deficit, and a score of 16 a more severe deficit). Group results for this measure are plotted in *Appendix 1—figure 1A* (left-most violin plot) and group comparisons between low ('Low ID' in green) and high ('High ID' in blue) mother's voice identification groups using Wilcoxon rank sum tests were not significant for ADOS Social scores (p = 0.83). In a second analysis, we examined whether mother's voice identification accuracy is related to social communication scores. Results from Pearson's correlation analysis indicates that mother's voice identification accuracy is not related to ADOS Social scores (*R* = 0.13, p = 0.59).

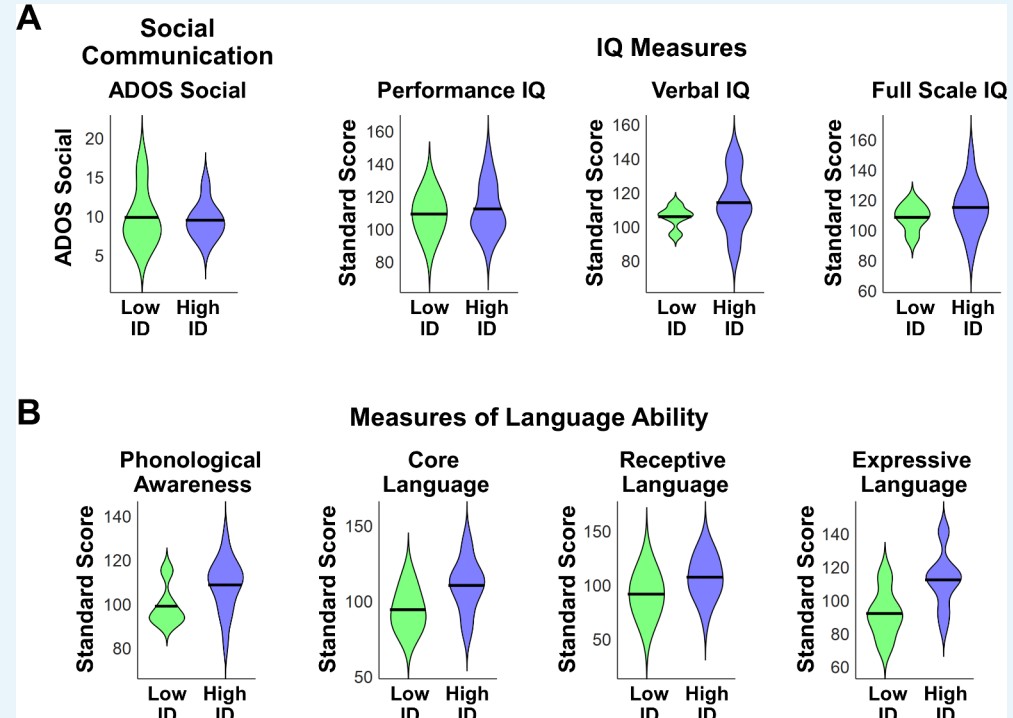

**Appendix 1—figure 1.** Social communication, cognitive, and language abilities in children with ASD with low vs. high mother's voice identification accuracy. (**A**) To examine whether children with ASD who were unable to identify their mother's voice in the mother's voice identification task (*N* = 5) showed a distinct behavioral profile relative to children with ASD who were able to perform this task (*N* = 16), we performed Wilcoxon rank sum tests using ADOS Social Affect scores (left-most violin plot) and standardized measures of IQ (Wechsler Abbreviated Scale of Intelligence (*Wechsler, 1999*)) between these groups. Group comparisons between low (green) and high (blue) mother's voice identification groups using Wilcoxon rank sum tests were not significant for social communication (p = 0.83) or IQ measures (p > 0.25 for all three measures, uncorrected for multiple comparisons). (**B**) To examine group differences in language abilities for low vs. high mother's voice identification groups, we performed Wilcoxon rank sum tests using CTOPP Phonological Awareness and CELF Language measures. Group comparison were not significant for any of the language measures (p > 0.05 for all four measures, not corrected for multiple comparisons), however there was a trend for reduced Core Language (p = 0.062) and Expressive Language abilities (p = 0.055) in the low (green) mother's voice identification group.
DOI: https://doi.org/10.7554/eLife.39906.011

We next examined whether there were any differences in standardized IQ scores (Wechsler Abbreviated Scale of Intelligence (*Wechsler, 1999*)) for children with low and high mother's voice identification accuracy, which are plotted below (*Appendix 1—figure 1A*, three right-most violin plots). Group comparisons between low and high mother's voice identification groups using Wilcoxon rank sum tests were not significant for any of the IQ measures

(p > 0.25 for all three measures, uncorrected for multiple comparisons). We then examined whether there were any differences for children with low vs. high mother's voice identification accuracy in standardized measures of language abilities, including CTOPP Phonological Awareness (*Wagner, 1999*) and CELF-4 Core Language, Receptive Language, and Expressive Language standard scores (*Semel, 2003*) (*Appendix 1—figure 1B*). Group comparison using Wilcoxon rank sum tests were not significant for any of the language measures (p > 0.05 for all four measures, not corrected for multiple comparisons), however there was a trend for reduced Core Language (p = 0.062) and Expressive Language abilities (p = 0.055) in the low (green) mother's voice identification group.

Together, results from clinical (i.e., social communication), cognitive, and language measures showed that there are no distinguishing features for the children with below chance mother's voice identification accuracy compared to children with above chance accuracy.

## Activation to unfamiliar female voices in TD children and children with ASD

We identified brain regions that showed increased activation in response to unfamiliar female voices compared to non-vocal environmental sounds separately within TD and ASD groups. This particular comparison has been used in studies examining the cortical basis of general vocal processing in neurotypical adult (*Belin et al., 2000*) and child listeners (*Abrams et al., 2016*). The TD child sample showed strong activation in bilateral superior temporal gyrus (STG) and sulcus (STS), amygdala, and right-hemisphere supramarginal gyrus of the inferior parietal lobule (IPL; *Appendix 1—figure 2A*). Children with ASD, however, showed a reduced activity profile in response to unfamiliar female voices, including a reduced extent of bilateral STG and STG and no difference in activity between unfamiliar female voices and environmental sounds in the amygdala (*Appendix 1—figure 2B*).

**Appendix 1—figure 2.** Brain activity in response to unfamiliar female voices compared to environmental sounds in TD children and children with ASD. (**A**) In TD children, unfamiliar female voices elicit greater activity throughout a wide extent of voice-selective superior temporal gyrus (STG) and superior temporal sulcus (STS), bilateral amygdala, and right-hemisphere supramarginal gyrus. (**B**) Children with ASD show a reduced activity profile in STG/STS in

response to unfamiliar female voices and do not show increased activity compared to environmental sounds in the amygdala.

DOI: https://doi.org/10.7554/eLife.39906.012

## Activation to mother's voice in TD children

We identified brain regions that showed greater activation in response to mother's voice compared to unfamiliar female voices separately within the TD and ASD groups. By subtracting out brain activation associated with hearing unfamiliar female voices producing the same nonsense words (i.e., controlling for low-level acoustical features, phoneme and word-level analysis, auditory attention), we estimated brain responses unique to hearing the maternal voice. TD children showed increased activity in a wide range of brain systems, including auditory, voice-selective, reward, social, and visual functions (*Appendix 1 —figure 3A*). Specifically, mother's voice elicited greater activation in primary auditory regions, including bilateral inferior colliculus (IC), the primary midbrain nucleus of the ascending auditory systems, and Heschl's gyrus (HG), which includes primary auditory cortex. Mother's voice also elicited greater activity in TD children in auditory association cortex in the superior temporal plane, including planum polare and planum temporale, with a slightly increased extent of activation in the right-hemisphere. Additionally, mother's voice elicited greater activity in a wide extent of bilateral voice-selective STS, extending from the posterior-most aspects of this structure ($y = -52$) to anterior STS bordering the temporal pole ($y = 6$). Preference for mother's voice was also evident in the medial temporal lobe, including left-hemisphere amygdala, a key node of the affective processing system, and bilateral posterior hippocampus, a critical structure for declarative and associative memory. Structures of the mesolimbic reward pathway also showed greater activity for mother's voice, including bilateral nucleus accumbens and ventral putamen in the ventral striatum, orbitofrontal cortex (OFC), and ventromedial prefrontal cortex (vmPFC). Mother's voice elicited greater activity in a key node of the default-mode network, instantiated in precuneus and posterior cingulate cortex, a brain system involved in processing self-referential thoughts. Preference for mother's voice was also evident in visual association cortex, including lingual and fusiform gyrus. Next, mother's voice elicited greater activity in bilateral anterior insula, a key node of the brain's salience network. Finally, preference for mother's voice was evident in frontoparietal regions, including right-hemisphere pars opercularis [Brodmann area (BA) 44] and pars triangularis (BA 45) of the inferior frontal gyrus, the angular and supramarginal gyri of inferior parietal lobule (IPL), and supplementary motor cortex.

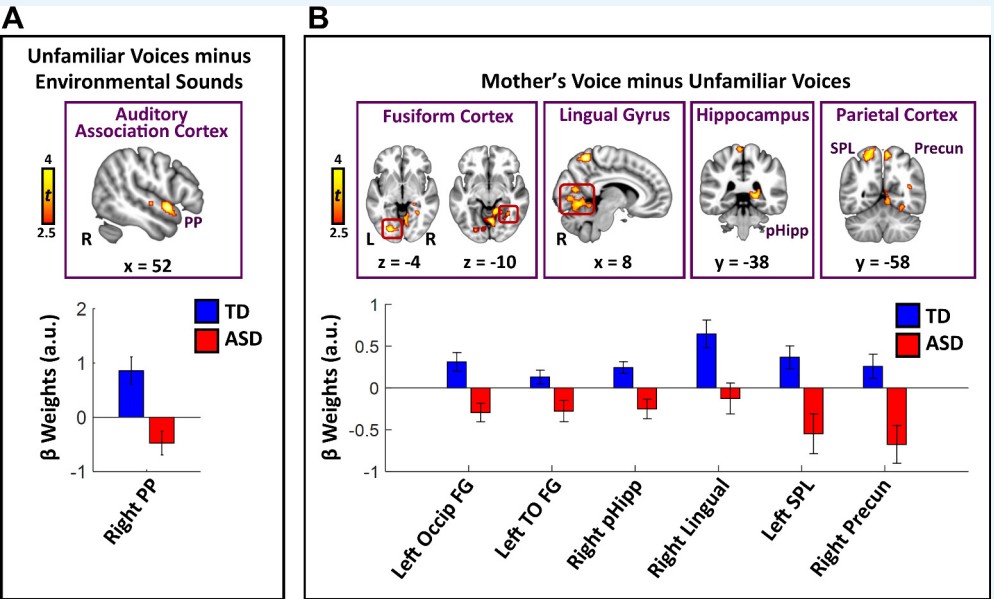

**Appendix 1—figure 3.** Signal levels in response to unfamiliar female voices and mother's voice in TD children and children with ASD. The reason for the signal level analysis is that stimulus-based differences in fMRI activity can result from a number of different factors. Significant differences were inherent to this ROI analysis as they are based on results from the whole-brain GLM analysis (*Vul et al., 2009*); however, results provide important information regarding the magnitude and sign of fMRI activity. (**a**) Regions were selected for signal level analysis based on their identification in the TD > ASD group difference map for the [unfamiliar female voices vs. environmental sounds] contrast (*Figure 2A*). ROIs are 5 mm spheres centered at the peak for these regions in the TD > ASD group difference map for the [unfamiliar female voices vs. environmental sounds] contrast. (**b**) Regions were selected for signal level analysis based on their identification in the [mother's voice vs. unfamiliar female voices] contrast (*Figure 2B*). The posterior hippocampus ROI is a 2 mm sphere centered at the peak for this regions in the [mother's voice >unfamiliar female voices] contrast. All other ROIs are 5 mm spheres centered at the peak for these regions in the TD > ASD group difference map for the [mother's voice vs. unfamiliar female voices] contrast.

DOI: https://doi.org/10.7554/eLife.39906.013

## Activation to mother's voice in children with ASD

Children with ASD showed a smaller collection of brain regions that were preferentially activated by mother's voice (*Appendix 1—figure 4B*). This group did not show a preference for mother's voice in primary auditory regions, including the IC, and activity in auditory cortex was confined to a small extent of left-hemisphere HG. Preference for mother's voice was also more limited in both auditory association cortex of the superior temporal plane as well as voice selective STS, particularly in the right hemisphere, where only a focal anterior STS (aSTS) cluster showed increased activity for mother's voice. Children with ASD also did not show a preference for mother's voice in medial temporal lobe structures, including both amygdala and hippocampus, as well as structures of the mesolimbic reward pathway, default mode network, and occipital regions. Children with ASD did, however, show increased activation to mother's voice in bilateral anterior insula of the salience network as well as frontoparietal regions, including left-hemisphere BA 44, bilateral supramarginal gyrus, and left-hemisphere angular gyrus.

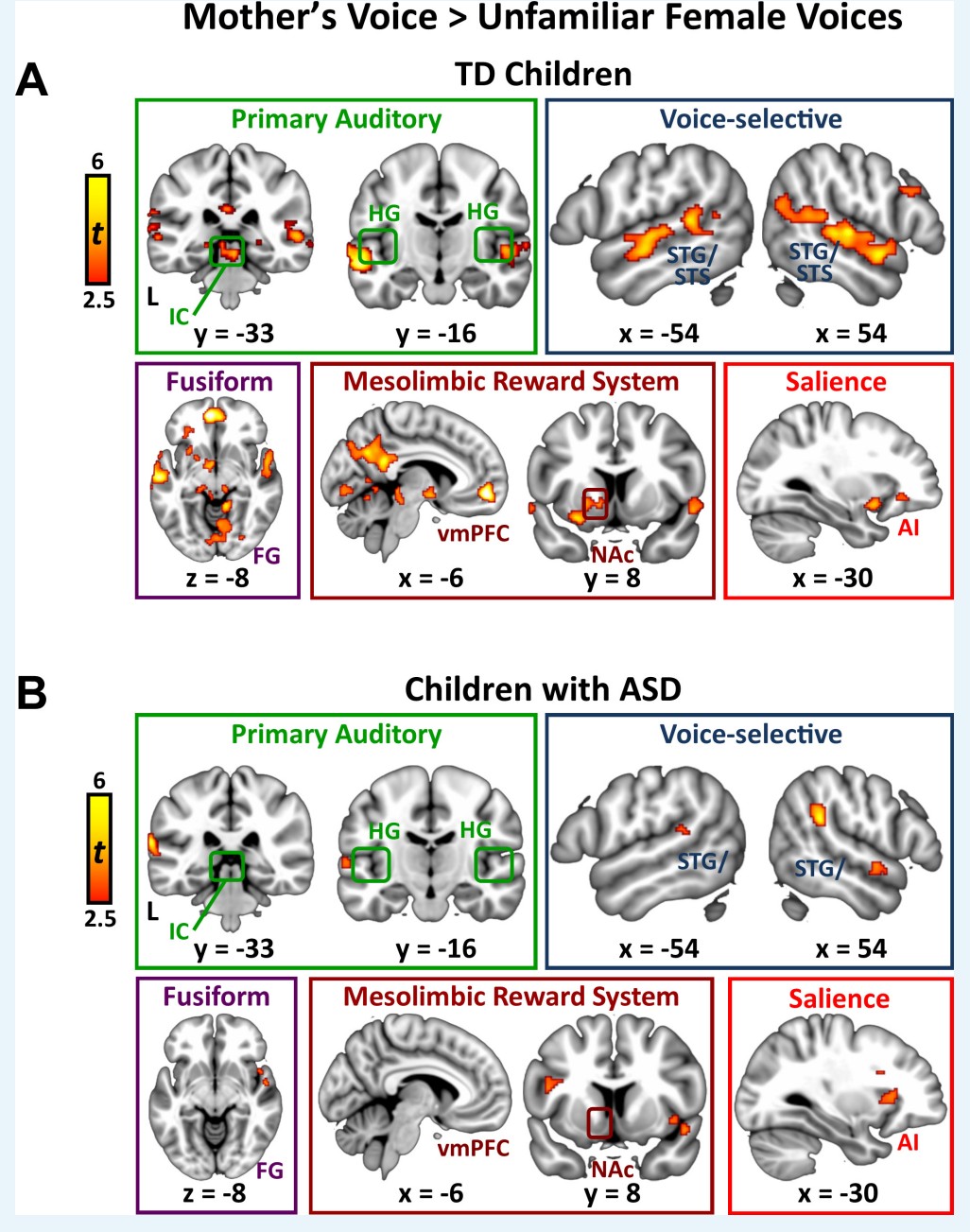

**Appendix 1—figure 4.** Brain activity in response to mother's voice compared to unfamiliar female voices in TD children and children with ASD. (**A**) In TD children, mother's voice elicited greater activity in auditory brain structures in the midbrain and superior temporal cortex (*top row, left*), including bilateral inferior colliculus (IC) and primary auditory cortex (medial Heschl's gyrus; mHG) and a wide extent of voice-selective superior temporal gyrus (STG; *top row, middle*) and superior temporal sulcus (STS). Mother's voice also showed greater activity in occipital cortex, including fusiform cortex (*bottom row, left*) as well as core structures of the mesolimbic reward system, including bilateral medial prefrontal cortex (mPFC) and nucleus accumbens (NAc), and the anterior insula (AI) of the salience network. (**B**) Greater activity for mother's voice was evident in a smaller collection of brain regions in children with ASD compared to TD children. Mother's voice did not elicit greater activity in auditory brain structures in the midbrain but extended slightly into primary auditory cortex (*top row, left*), and activated a more limited extent of voice-selective STG (*top row, middle*) and STS. Mother's voice did not elicit greater activity compared to unfamiliar female voices in fusiform

cortex, and mesolimbic reward system. Mother's voice did elicit greater activity in AI of the salience network.

DOI: https://doi.org/10.7554/eLife.39906.014

## fMRI activation and connectivity profiles in children with ASD are not related to mother's voice identification accuracy

Behavioral results indicated that 5 of the 21 children with ASD had below chance-level accuracy on the mother's voice identification task (*Figure 1C*; see Results, Identification of Mother's Voice). An important question is whether the five children with ASD who performed below chance on the mother's voice identification task might show a distinct neural signature that may help explain why these children were unable to identify their mother's voice in our behavioral task. A plausible hypothesis is that the five children who were unable to identify their mother's voice in the task would show reduced brain activation in response to unfamiliar or mother's voice stimuli. To test this hypothesis, we performed additional analyses to examine whether there are any identifying neural characteristics regarding these five children with low identification accuracy.

We first examined neural response profiles for the five children with low vs. high mother's voice identification accuracy by plotting ROI signal levels for the contrasts and regions identified in *Figure 3A*. First, results showed no group differences between children with low vs. high identification accuracy using Wilcoxon rank sum tests for any of the brain regions associated with the [unfamiliar voices vs. non-social environmental sounds contrast] (*Appendix 1—figure 5A*; p > 0.35 for all three regions, not corrected for multiple comparisons). We then examined low vs. high identification accuracy using Wilcoxon rank sum tests for the brain regions associated with the [mother's voice vs. unfamiliar voices contrast] (*Figure 3B*) and again found no group differences (*Appendix 1—fFigure 5B*; p > 0.45 for all seven regions, not corrected for multiple comparisons).

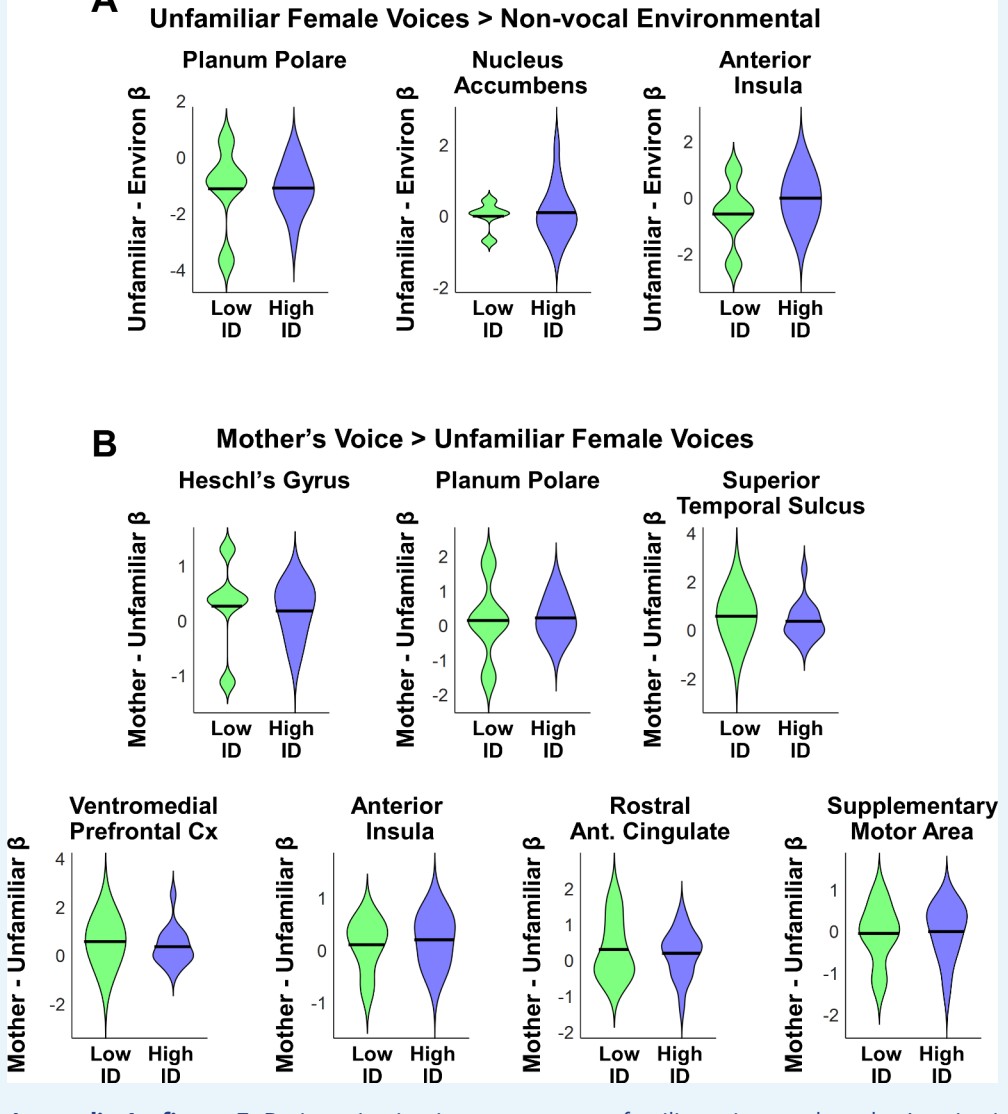

**Appendix 1—figure 5.** Brain activation in response to unfamiliar voices and mother's voice in children with ASD with low vs. high mother's voice identification accuracy. (**A**) To examine whether children with ASD who were unable to identify their mother's voice in the mother's voice identification task (*N* = 5) showed a distinct neural response profile relative to children with ASD who were able to perform this task (*N* = 16), Wilcoxon rank sum tests were computed using ROI single levels (mean contrast betas) for the [unfamiliar voices minus non-social environmental sounds] in regions identified in *Figure 3A*. Results showed no group differences between children with low (green) vs. high (blue) identification accuracy for any of the brain regions associated with the [unfamiliar voices vs. non-social environmental sounds] contrast (p > 0.35 for all three regions, not corrected for multiple comparisons). (**B**) Group differences in neural response profiles for low vs. high mother's voice identification groups using ROI single levels (mean contrast betas) for the [mother's voice minus unfamiliar voices] contrast were computed within regions identified in *Figure 3B*. Results showed no group differences between children with low vs. high identification accuracy for any of the brain regions associated with the [mother's voice minus unfamiliar voices] contrast (p > 0.45 for all seven regions, not corrected for multiple comparisons).

DOI: https://doi.org/10.7554/eLife.39906.015

We examined whether mother's voice identification accuracy affected results from ADOS covariate analyses in children with ASD (*Figure 3*). Therefore, additional regression analyses

were performed in which ADOS Social Affect scores were the dependent variable and predictors included mother's voice identification accuracy and betas from ROIs identified in the [unfamiliar female voice minus environmental sounds] contrast (i.e., *Figure 3A*) or [mother's voice minus unfamiliar voices] contrast (i.e., *Figure 3B*). Separate regression models were computed for each ROI in each vocal contrast. Results showed that all ROI signal levels reported in *Figure 3* were significant predictors of social communication scores after regressing out mother's voice identification accuracy ($p \leq 0.005$ for all ROIs).

We then examined whether removing the five children with low mother's voice identification accuracy would affect group GLM and functional connectivity results. We therefore examined a sub-group comprised of the 16 children with ASD who showed above chance identification accuracy and performed whole-brain TD vs. ASD group comparisons, social communication covariate analysis within the ASD group, and functional connectivity analyses, including SVC and SVR. Results for all analyses were similar to those described previously for the entire ASD group. Specifically, whole-brain TD vs. ASD group differences and social communication covariate results were evident in similar brain regions as those described for the larger ASD group. Functional connectivity results also showed the same pattern of results described for the entire ASD group: SVC results showed that connectivity during unfamiliar voice processing could not classify individuals with ASD from TD children (SVC Accuracy = 50.9%, p = 0.41) while connectivity during mother's voice processing could classify individuals with ASD from TD children (SVC Accuracy = 66.3%, p = 0.014). Furthermore, SVR results showed that connectivity using combined features from both unfamiliar and mother's voice processing could classify individuals with ASD from TD children ($R$ = 66.3%, p = 0.003). These results indicate that patterns of brain activity and connectivity in children with ASD in response to vocal stimuli were unrelated to behavioral identification of mother's voice.

Together, results from neural measures of voice processing showed that there are no distinguishing features for the children with below chance mother's voice identification accuracy compared to children with above chance accuracy.

**Appendix 1—table 1.** Effect sizes for GLM results: TD vs. ASD Group Analysis. The overall effect size measured across all brain clusters identified in the TD vs. ASD Group Analyses is 0.68.

| Contrast | Brain region | Effect size |
|---|---|---|
| [Unfamiliar Voices minus Environmental Sounds] | Right-hemisphere Planum Polare (PP) | 0.70 |
| [Mother's Voice minus Unfamiliar Voices] | Right-hemisphere Intercalcarine | 0.65 |
| | Right-hemisphere Lingual | 0.68 |
| | Right-hemisphere Fusiform | 0.66 |
| | Left-hemisphere Fusiform | 0.67 |
| | Right-hemisphere Hippocampus | 0.66 |
| | Left-hemisphere Superior Parietal Lobule (SPL) | 0.69 |
| | Right -hemisphere Precuneus | 0.69 |

DOI: https://doi.org/10.7554/eLife.39906.016

**Appendix 1—table 2.** Effect sizes for GLM results: Social Communication Covariate Analysis. The overall effect size measured across all brain clusters identified in the Social Communication Covariate Analysis is 0.76.

| Contrast | Brain region | Effect size |
|---|---|---|

*Appendix 1—table 2 continued on next page*

*Appendix 1—table 2 continued*

| Contrast | Brain region | Effect size |
|---|---|---|
| [Unfamiliar Voices minus Environmental Sounds] | Right-hemisphere Planum Polare (PP) | 0.84 |
| | Left-hemisphere Nucleus Accumbens (NAc) | 0.69 |
| | Right-hemisphere Anterior Insula (AI) | 0.84 |
| [Mother's Voice minus Unfamiliar Voices] | Left-hemisphere Heschl's Gyrus (HG) | 0.77 |
| | Left-hemisphere Planum Polare (PP) | 0.77 |
| | Right-hemisphere Superior Temporal Sulcus (mSTS) | 0.74 |
| | Right-hemisphere Ventromedial prefrontal cortex (vmPFC) | 0.73 |
| | Left-hemisphere Anterior Insula (AI) | 0.77 |
| | Right-hemisphere Rostral Antreior Cingulate Cortex (rACC) | 0.73 |
| | Right-hemisphere Supplementary Motor Area (SMA) | 0.76 |

DOI: https://doi.org/10.7554/eLife.39906.017

**Appendix 1—table 3.** Brain regions used in functional connectivity analyses.

| Brain region | Coordinates |
|---|---|
| Left-hemisphere pSTS | [−63–42 9] |
| Right-hemisphere pSTS | [57 -31 5] |
| Left-hemisphere vmPFC | [−6 32–14] |
| Right-hemisphere vmPFC | [6 54 -4] |
| Left-hemisphere Anterior Insula | [−28 18–10] |
| Right-hemisphere VTA | [2 -22 -20] |
| Left-hemisphere NAc | [−12 18–8] |
| Right-hemisphere NAc | [14 18 -8] |
| Left-hemisphere OFC | [−36 24–14] |
| Left-hemisphere Putamen | [−24 14–8] |
| Right-hemisphere Putamen | [16 14 -10] |
| Left-hemisphere Caudate | [−18 4 20] |
| Right-hemisphere Caudate | [14 22 -6] |
| Right-hemisphere Amygdala | [30 -4 -24] |
| Right-hemisphere Hippocampus | [28 -6 -26] |
| Right-hemisphere Fusiform | [36 -28 -22] |

DOI: https://doi.org/10.7554/eLife.39906.018

**Appendix 1—table 4.** GLM Threshold Analysis: TD vs. ASD Group Analysis [Unfamiliar Voices minus Environmental Sounds] fMRI Contrast.

| Brain Region Activation | Height: p<0.005 Extent: p<0.05 | Height: p<0.005 Extent: p<0.01 | Height: p<0.001 Extent: p<0.05 | Height: p<0.001 Extent: p<0.01 |
|---|---|---|---|---|
| | *67 Voxels* | *87 Voxels* | *30 Voxels* | *41 Voxels* |
| Auditory Assoc. Cx, PP | Yes | Yes | Yes | Yes |

DOI: https://doi.org/10.7554/eLife.39906.019

**Appendix 1—table 5.** GLM Threshold Analysis: TD vs. ASD Group Analysis [Mother's Voice minus Unfamiliar Voices] contrast.

| Brain Region Activation | Height: p<0.005 Extent: p<0.05 | Height: p<0.005 Extent: p<0.01 | Height: p<0.001 Extent: p<0.05 | Height: p<0.001 Extent: p<0.01 |
|---|---|---|---|---|
| | 67 Voxels | 87 Voxels | 30 Voxels | 41 Voxels |
| Occipital Fusiform Gyrus | Yes | Yes | Yes | No |
| Temporal Occipital Fusiform Gyrus | Yes | Yes | No | No |
| Post. Hippocampus | Yes | Yes | No | No |
| Lingual Gyrus | Yes | Yes | Yes | Yes |
| Superior Parietal | Yes | Yes | Yes | Yes |
| Precuneus | Yes | Yes | Yes | Yes |

DOI: https://doi.org/10.7554/eLife.39906.020

**Appendix 1—table 6.** GLM Threshold Analysis: Social Communication Covariate Analysis, [Unfamiliar Voices minus Environmental Sounds] fMRI Contrast.

| Brain Region Activation | Height: p<0.005 Extent: p<0.05 | Height: p<0.005 Extent: p<0.01 | Height: p<0.001 Extent: p<0.05 | Height: p<0.001 Extent: p<0.01 |
|---|---|---|---|---|
| | 67 Voxels | 87 Voxels | 30 Voxels | 41 Voxels |
| Auditory Assoc., PP | Yes | Yes | Yes | Yes |
| Voice Selective, STG | Yes | Yes | Yes | Yes |
| Mesolimbic Reward, NAc | Yes (SVC) | No | No | No |
| Salience, AI | Yes | Yes | Yes | Yes |

DOI: https://doi.org/10.7554/eLife.39906.021

**Appendix 1—table 7.** GLM Threshold Analysis: Social Communication Covariate Analysis, [Mother's Voice minus Unfamiliar Voices] fMRI Contrast.

| Brain Region Activation | Height: p<0.005 Extent: p<0.05 | Height: p<0.005 Extent: p<0.01 | Height: p<0.001 Extent: p<0.05 | Height: p<0.001 Extent: p<0.01 |
|---|---|---|---|---|
| | 67 Voxels | 87 Voxels | 30 Voxels | 41 Voxels |
| Primary Auditory, HG | Yes | Yes | Yes | Yes |
| Voice-selective, STG/STS | Yes | Yes | No | No |
| Mesolimbic Reward, vmPFC | Yes | Yes | Yes | Yes |
| Salience, AI | Yes | Yes | Yes | Yes |
| Salience, rACC | Yes | Yes | No | No |
| Motor, SMA | Yes | Yes | Yes | Yes |

DOI: https://doi.org/10.7554/eLife.39906.022

