## [Decision Letter]

[**Editorial note:** This article has been through an editorial process in which the authors decide how to respond to the issues raised during peer review. The Reviewing Editor's assessment is that minor issues remain unresolved.]

Re-evaluation:

We appreciate that you have highlighted the sample size as a limitation, provided documentary support for the sample size, your deeper individual testing, and noted the need for future studies that are better able to capture the heterogeneity of ASD. Nonetheless, there is increased sensitivity to the (out-of-sample) reproducibility issues inherent in studies of this size, even given the challenges of clinical research of this note. While we support publication of the paper, this limitation has been raised in the peer review process.

Decision letter after peer review:

Thank you for submitting your article "Impaired voice processing in reward and salience circuits predicts social communication in children with autism" for consideration by *eLife*. Your article has been reviewed by three peer reviewers, one of whom is a member of our Board of Reviewing Editors, and the evaluation has been overseen by Michael Frank as the Senior Editor. The following individual involved in review of your submission has agreed to reveal her identity: Coralie Chevallier (Reviewer #1).

The Reviewing Editor has highlighted the concerns that require revision and/or responses, and we have included the separate reviews below for your consideration. If you have any questions, please do not hesitate to contact us.

This is a very interesting and nicely presented study of cortical responses to mothers' versus strangers' voices on people with Autistic Spectrum Disorders. All reviewers felt the study was well designed in its stimuli and, in principle, impressed by the range of uni- and multivariate analyses undertaken.

However, all three reviewers raise substantial concerns regarding the small sample size: these reflect the ability of the study to provide adequate cover of the heterogeneity of ASD, the possibility that some effects may be inflated (particularly the regressions), the under-estimate of variance in the cross-validation tests and, most importantly, the likely challenges of reproducing the principle findings (in the absence of an independent test data set). The regression effect sizes are very strong, and there is possibly some colinearity between the test to identify the ROIs (the group contrast) and the subsequent regression (ASD severity), given that ASD typically presents as an extreme on the healthy development/social spectrum.

Each analysis, on its own, may be valid, but the general feeling is that the authors are doing too much with this limited data set. In the absence of acquiring further data, there seems to be little additional work that the authors can do to address this fundamental concern.

Given this is part of the *eLife* "trial", there does not seem to be a deep enough concern for the paper to be withdrawn from further consideration. However, if published, the important concerns of the reviewers will need to be published alongside the paper, pending the nature of the authors' responses.

Other concerns are less fundamental and are listed below.

Separate reviews (please respond to each point):

*Reviewer #1:*

Overall, I very much enjoyed reading the paper: clearly written, timely, and novel. The use of recordings that are specific to each participant is an important contribution. I am not a neuroscientist so I am not qualified to evaluate the quality of the neuroimaging work. I will therefore focus my remarks on the Introduction, clinical / behavioural aspects of the Materials and methods, and Discussion. (and please pardon my naivety when I do comment on the neuroimaging part of the work).

1) Behavioural tasks can provide mechanistic insights

I think the authors should tone down their claim about the limitations of behavioural studies to understand and tease apart different cognitive mechanisms. Many behavioural studies use ingenious paradigms to tease apart various mechanistic possibilities. Recently, the use of computational modeling methods applied to behavioural data has also been very fruitful in this respect (different model parameter reflect different mechanisms)

On the same topic, in the Introduction “…and obtaining valid behavioral measurements regarding individuals’ implicit judgments of subjective reward value of these stimuli can be problematic”: I was surprised by the strong statement. Antonia Hamilton has published many papers demonstrating the validity of behavioural tools to measure social reward responsiveness. I have also published several papers on that same topic (including one using signal detection theory, in PLOS One). In these papers, no abstract judgment is required. Rather, participants' behaviours are thought to reflect underlying social motivation or social reward responsiveness.

2) Why is the accuracy for mother's voice identification different in the ASD vs. TD group?

I would like to know a bit more the difference in accuracy detection between the groups: why did the ASD group perform below the TD group? The authors point out that 5 children performed below chance in identifying their own mother's voice. Is there evidence that this indicates a true deficit or is poor performance linked to other factors (poor hearing? deteriorated listening skills in the scanner?). Where are these five children on the regression? Do they drive the effect?

If some children did not recognize their mother's voice, it seems to be that they should be looked at differently: if the reward / memory / visual network is less activated in these children, is it because they do not find voices as rewarding / memorable or is it because they didn't recognize this familiar voice (but if they had, their brain would have reacted in the same way)?

Another naive question on the same question. The authors report that "fMRI activation profiles in children with ASD were not related to mother's voice identification accuracy". So I was left wondering what these activations reflect (if they are not sensitive to the fact that 5 out of 21 children did not identify their mother, does it mean that these activations are picking up on something that is much more domain general than anything that might have to do with "mother's voice" specifically?) Is there a way to statistically correct all analyses for accuracy levels?

The authors should report whether identification accuracy is related to ADOS SC scores.

3) ADOS score use

ADOS scores are not meant to be used as a continuous severity scores unless they are transformed (Gotham, K., Pickles, A., & Lord, C. (2009). Standardizing ADOS scores for a measure of severity in autism spectrum disorders. Journal of autism and developmental disorders, 39(5), 693-705.). I do not know how this logic applies to using the social affect score only but it should be checked / discussed.

4) Are there a priori criteria for exclusion based on motion?

Materials and methods subsections “Participants” and “Movement criteria for inclusion in fMRI analysis”: Is there an a priori threshold to exclude participants based on motion? Or published guidelines? Can the authors specify the exclusion decision criteria? Is there a group difference in average motion?

5) Sample size concerns

In a behavioural study, a sample size of 21 is now considered too small. I realise that the change in standard is recent (and I have published many papers using small sample sizes myself). However, I do think that we need to accelerate change, especially for conditions that are notoriously heterogenous, such as ASD. And especially when one is interested in explaining interindividual differences (by using correlations).

Reviewers who specialise in neuroimaging methods may want to comment on this specific point but the paper would definitely be stronger is the sample size exceeded (or at least matched) the current average in cognitive neuroscience field (ie 30, see Poldrack et al., 2015, Figure 1). Alternatively, the authors should report observed power. This is I think most important for the regression part of the paper. If power is too low, I would recommend moving this part of the paper to the SM and rewriting the main text accordingly.

Button, K. S., Ioannidis, J. P., Mokrysz, C., Nosek, B. A., Flint, J., Robinson, E. S., & Munafò, M. R. (2013). Power failure: why small sample size undermines the reliability of neuroscience. Nature Reviews Neuroscience, 14(5), 365.

Poldrack, Russell A., et al. "Scanning the horizon: towards transparent and reproducible neuroimaging research." Nature Reviews Neuroscience 18.2 (2017): 115.

To sum up, I think the paper raises an interesting question and would be even better if reproducibility concerns were addressed by increasing sample size.

*Reviewer #2:*

Impaired voice processing in.… children with autism" by Abrams et al.

This paper reports the uni- and multivariate analysis of fMRI acquired from children with ASD while listening to their mothers' or strangers' voice. The authors find quite striking correlations between social disability rating scales and both brain activity and functional connectivity.

The experimental stimuli, writing, analysis and cohort characterization are all of a very high standard. There are some limitations in the actual design that limit the nature of the inferences drawn. In a revised manuscript, these should be addressed through a subtle reframing and possibly additional discussion points.

1) The task stimuli are very well controlled but the task is essentially a passive listening task, with a low level vigilance task designed simply to keep the participants engaged with the stimuli. Therefore, in the absence of an explicit reward, the authors should be cautious of falling foul of reverse inference (Poldrack, 2006) when framing the findings in terms of "reward circuits". I guess listening to a voice, particularly a mother's voice, is implicitly rewarding, but caution should be taken in drawing to direct an inference in regards to dysfunctional reward processes (see Discussion section). Indeed, the whole positioning of functional neuroimaging studies as being more informative than behavioural tests should be mindful of the limitations of inferring disrupted functional circuits in disorders unless you are actually probing that function with an appropriate task.

2) Similarly, there is no manipulation of attention and therefore no means of telling whether the differences in activation during voice perception in ASD is simply a lack of due interest in, and attention to voices preceding (or consequent) to changes in reward-based learning.

3) Again the authors draw a very direct line between mother's vs stranger's voice and social reasoning/communication (e.g. Abstract). But maternal voice is more than a simple cue but incorporates many other processes, including basic parental attachment and dyadic reciprocation.

4) What are the core defining disturbances underlying the diagnosis of ASD in this study? The information in paragraph two of subsection “Participants” simply refers to an algorithm. While this might be sufficient for reproducibility, it is inadequate here for two reasons: First, as *eLife* is a general (not a clinical) journal, more depth and context is required for the broader readership. Second, if the diagnosis relies heavily on social deficits (which I suspect it does), is it then surprising that the strength of between group differences covaries so markedly within the ASD with social deficit scores. If so, is there an implicit circularity between the identification of these regions (Figure 2) and the very strong correlations against ADOS scores? If not, what other effects may be driving these? Such strong correlations are bound to draw attention and I think the authors should therefore pay careful attention to this issue.

5) This is a modest sample size for the use of machine learning, particularly when using a high dimensional (functional connectivity) feature space. N-fold cross-validation does always not provide strong control here (Varoquaux, 2017): Do the authors undertake a feature reduction step, such as a LASSO? Is there some logical circularity between identifying features with a group contrast, then using these same features in a between group classifier? Also, I would avoid using the term "prediction" for contemporaneous variables, even when cross-validation is undertaken.

6) I don't see any model-based analysis of neuronal interactions and hence don't think the authors are examining effective connectivity. gPPI is a purely linear model of statistical dependences and their moderation and hence falls into the class of functional connectivity. Personally, I would prefer to have seen the author use a more dynamic, model-driven method of effective connectivity, using something like DCM to provide a deeper mechanistic insight into the changes in activation and information flow (I also am not sure I buy into the choice of ROI's that do not show the group effects). However, given the classification success, the authors' approach seems entirely reasonable. However, please do specify the nature of the gPPI model in the text and supplementary material (what are the nodes, inputs and modulators; is it possible to represent this graphically?).

7) Do the manipulations to the vocal signals (subsection “Stimulus post-processing”) possibly warp the sound of the speech? If so, could this influence the ASD responses (ASD being perhaps more tuned to low level features of stimulus inputs).

References:

Poldrack, R. A. (2006). Can cognitive processes be inferred from neuroimaging data? Trends in cognitive sciences, 10(2), 59-63.

Varoquaux, G. (2017). Cross-validation failure: small sample sizes lead to large error bars. Neuroimage.

Minor Comments:

1) Introduction, second sentence: "affected" -> "ASD"

2) Figure 1B caption: Please add stats

3) Discussion first paragraph: "These findings.…"; Third paragraph: "Our findings.…"

4) Discussion paragraph three: ?"predicts"?

5) Subsection “Participants”: please add more details regarding the diagnosis; e.g. "In essence, these ASD.…" (see point 4 above)

6) Table 1: What are the "typical" values of ADOS-social and ADI-A social"? Are these very impaired children (also see point 4 above).

7) Materials and methods: Were there any between group differences in head motion parameters?

8) Subsection “Effective Connectivity Analysis”: I'm not sure what "preempts.… biases" means here – the authors are not modelling effects in the present data, which I think is a shame (see point 6 above)

*Reviewer #3:*

In this study, Abrams and colleagues examined voice processing in children (average age 10 years old) with and without autism. The specifically were interested in reward and salience circuitry and relate the study back to predictions made by the social motivation theory of autism. The authors applied group-level analyses, brain-behavior correlation analyses, as well as PPI connectivity analyses and applied multivariate classifier and regression analyses applied to the connectivity data. There are several issues which the authors may want to address.

1) Small sample size. The sample size for the study was n=21 per group. This sample size is likely not large enough to cover substantial heterogeneity that exists across the population of individuals with autism diagnosis. Thus, questions about generalizability arise. Can future studies replicate these findings? Small sample size also means lower statistical power for identifying more subtle effects, and this is especially important for the context of whole-brain between-group or brain-behavior correlation analysis which the authors have solely relied on for the activation and clinical correlation analyses (see Cremers, Wager, & Yarkoni, 2017, PLoS One). For multivariate classifiers and regressions, these too produce inflated and over-optimistic levels of predictions with smaller sample size (e.g., Woo et al., 2017, Nature Neuroscience).

2) Given the small sample sizes, but relatively strong and justified anatomical hypotheses, why not run ROI analyses instead of whole-brain analyses? Statistical power would likely be increased for ROI analyses, and one can cite more unbiased estimates of effect size. Whole-brain analyses can show us is where the likely effects might be, and this is helpful when we don't have strong anatomical hypotheses. But here the authors do have strong anatomical hypotheses, yet they choose an analysis approach that is not congruent with that and penalizes them in terms of statistical power and doesn't allow for estimation of unbiased effect sizes. What is missing from the paper is an estimate of how big the effects are likely to be, as this is what we should ideally care about (Reddan, Lindquist, & Wager, 2017, JAMA Psychiatry). Future studies that may try to replicate this study will need to know what the effect sizes are likely to be. Meta-analyses ideally need unbiased estimates of effect size. However, all we have to go off of here are the authors figures showing whole-brain maps, that likely just tell us where some of the largest effects may likely be.

3) Reported effect sizes in Figure 3 (e.g., Pearson's r) are likely inflated given small sample sizes and also due to the fact that it appears that the reported r values in the figure are likely taken from the peak voxel.

4) Scatterplots in Figure 3 show inverted y-axes so that higher numbers on at the bottom and lower numbers are at the top. The reported correlations are negative, and yet the scatterplot shows what looks like a positive correlation. All this confusion is due to the inverted y-axes. The authors should correct the plots to avoid this confusion.

5) The authors heavily rely on reverse inference to relate their findings back to the social motivation theory. However, if their manipulations were powerful enough to create a distinction between a stimulus that was heavily socially rewarding (e.g., mother's voice) versus another that is not (e.g., unfamiliar voice), then shouldn't there be some kind of difference in the main activation analysis in reward-related areas (i.e. Figure 2B)? In other words the main contrast of interest that might have been most relevant to the social motivation theory produces no group differences in activation in areas like the ventral striatum. Because this contrast doesn't really pan out the way the theory predicts, doesn't this cast doubt on the social motivation theory, or couldn't it be that some of the contrasts in this study can be better explained by some other kind of reverse inference than the social motivation theory?

6) From what I could tell, no manipulation check was done to measure some aspect of how rewarding the stimuli were to participants. This seems critical if the authors want to make strong reverse inferences back to the social motivation theory.

7) The authors should include a limitations section to their paper.

[Editors' note: further revisions were suggested prior to acceptance, as described below.]

Thank you for submitting your revised article "Impaired voice processing in reward and salience circuits predicts social communication in children with autism" for consideration by *eLife*. Your article has been reviewed by two peer reviewers, and the evaluation has been overseen by a Reviewing Editor and Michael Frank as the Senior Editor. The following individual involved in review of your submission has agreed to reveal her identity: Coralie Chevallier (Reviewer #1).

The reviewers find that, on the whole, you have responded to the technical concerns raised on the prior round.

Reviewer #3 remains concerned that the sample size is not sufficiently large to capture the heterogeneity of ASD. The reviewer, BRE and senior editor have corresponded and agree that this is an important limitation regarding the broader generalizability of the study. In other words, even though you have adequately controlled for in-sample error, generalization beyond the study sample is limited by the size of the cohort.

We feel that a stronger statement that acknowledges this should be added to the manuscript. Given the advanced stage of the manuscript, we recognize that acquiring further subjects is highly unlikely to be feasible although we will note, in the published reviews, that this remained of concern at the conclusion of the reviewing process.

The paper will be promptly be re-assessed by the Reviewing Editor upon resubmission.

*Reviewer #1:*

I would like to thank the authors for their extremely thorough response and for all the additional analyses they performed in response to my comments. The authors have suitably addressed all of my comments and incorporated a limitations section to mention the modest sample size. Thank you again for this detailed and careful response.

*Reviewer #3:*

In this reviewer's opinion, the author's responses to previous comments largely dismissed many of the key issues in the comments, and the arguments brought to counter those comments were not at all convincing. The authors need to more properly consider the issues and at the very least comment on them as potential drawbacks, limitations, caveats, etc.

For example, none of the author's responses are adequate for addressing the main problem of small sample size. Heterogeneity across the autism population is vast and one cannot likely cover much of it with an n=21. Discussion about symptom domains as 'constraining' heterogeneity are not relevant to this aspect of the issue. If another study comes up with similar paradigm, but also small n, how likely will that study replicate the findings of the current study? It depends. If the new study samples a different strata of the autism population, then it may not replicate. Studies with larger sample size are more generalizable because they can cover a better range of the population and are not as susceptible to sampling biases that are more pronounced with small samples.

In another example, the authors claim that within subject, the multiple runs somehow can counteract the issue of small sample size. More data within subject can have an impact on statistical power. The estimates for each subject are more precise with more data within-subject. However, generalizing between-subjects or between studies is still an issue, for the reasons stated above.

With regard to the comment about ROI analysis, an analysis with 16 regions is surely more highly powered than a whole-brain analysis with 20,000 voxels to correct for. For anatomically constrained hypotheses (which the authors have), the comment still stands that it is strange to take a whole-brain approach as if the authors had a less strong idea about the anatomical hypotheses. Whole-brain analyses are lower in statistical power and have the ability to over-inflate effect sizes, and the authors are merely capitalizing on those things with small sample size.

Minor Comments:

The scatterplots must be plotted in the correct way, and not reversed, as this will mainly confuse readers.

---

## [Author Response]

This is a very interesting and nicely presented study of cortical responses to mothers' versus strangers' voices on people with Autistic Spectrum Disorders. All reviewers felt the study was well designed in its stimuli and, *in principle*, impressed by the range of uni- and multivariate analyses undertaken.However, all three reviewers raise substantial concerns regarding the small sample size: these reflect the ability of the study to provide adequate cover of the heterogeneity of ASD,

We thank the editors and reviewers for this comment. The ability of our sample to adequate cover heterogeneity in ASD is an important consideration for this study and all studies investigating the perceptual, cognitive, and biological bases of ASD. From one perspective, adequately addressing heterogeneity in ASD is a formidable challenge: studying and working with individuals with ASD has lead more than one researcher and clinician to state: “If you’ve met one person with autism, you’ve met one person with autism”. Given this widely-held view, sufficiently addressing heterogeneity in ASD represents a significant research challenge, particularly for pediatric brain imaging research in which gathering high-quality data in a modest sample such as that reported here can take many years.

Here we have addressed this challenge with a comprehensive approach which both: (a) constrains heterogeneity in ASD to a critical diagnostic symptom domain, social communication abilities, and (b) explores heterogeneity within this symptom domain by identifying neural features that covary as a function of social communication abilities. The rationale for constraining heterogeneity in ASD to a symptom domain is that all individuals with ASD necessarily have pronounced impairments in diagnostic domains, and these domains have considerably less heterogeneity in individuals with ASD relative to other behavioral and cognitive domains. For example, in the social communication domain, which we investigate in the current study, all individuals with ASD show pronounced deficits that are categorized by impaired communication and/or reciprocal social interactions (1). In contrast, consider the case of language function and IQ in children with ASD. Language function and IQ in these individuals varies widely, from very high levels, which are commensurate with neurotypical individuals, through very low levels of language and cognitive function (1-3). We argue that constraining analyses to an ASD diagnostic domain is an important approach for keeping a focus on critical areas that are central to the core deficits in the disorder, thereby providing important insight to clinicians, researchers, parents, and educators regarding the neurobiological bases of these core deficits.

The rationale for exploring heterogeneity within the social communication symptom domain is that high-functioning children with ASD, such as those included in our sample, often show a range of social communication abilities, from mild/moderate deficits to more severe deficits. For example, the Autism Diagnostic Observation Schedule (ADOS-2) is the gold-standard diagnostic instrument for ASD (4), and social communication subscores on the ADOS range from 0 to 22, with a 0 indicating no social communication deficit, a 7 indicating a more mild deficit, and a 22 indicating the most severe deficit. The high-functioning children included in our sample had a range of social communication abilities as measured with the ADOS between 7 and 16. Children with ASD who have scored higher than 16 tend to have multiple comorbid symptoms, and low cognitive function, thus confounding neuroscientific interpretation of findings. Importantly, understanding the link between social communication symptom severity and social brain processing is a critical question for understanding the neurobiological basis of autism, and is a question that has not been explored in previous studies of human voice processing in children with ASD. The reason this is an important question is that the severity of social communication deficits can play a crucial role in autism models. For example, the social motivation theory of ASD posits that impairments in representing the reward value of human vocal sounds impedes individuals with ASD from engaging with these stimuli and contributes to social interaction difficulties (5, 6). Given the causal link proposed in this model, a key prediction of the social motivation theory is that children with more severe deficits associated with social reward will have greater social communication deficits compared to those children with less severe social reward deficits. Therefore, by examining heterogeneity within the social communication symptom domain, we are able to test an important prediction of an influential ASD model and understand the neural features that may contribute to this heterogeneity.

In our study, we have employed this approach and constrained heterogeneity in children with ASD by focusing on social communication abilities. We then examine heterogeneity within the social communication symptom domain and show that social communication abilities explain significant variance in activity and connectivity of social reward and salience brain regions during human voice processing.

The possibility that some effects may be inflated (particularly the regressions),

We thank the reviewer for this comment. As we stated in the initial submission (see Brain Activity Levels and Prediction of Social Function subsection of the Materials and methods), the goal for performing the regression analyses was to examine the robustness and reliability of brain activity levels for predicting SC scores. The rationale for this analysis is that significant whole-brain covariate analysis can be the result of outliers in the data or other spurious effects (7). By extracting signal levels from ROIs, plotting the distributions, and performing additional regression analyses on these data, our goal was to show that the whole-brain covariate analysis was not driven by outlying data and was robust to confirmatory cross-validation regression analysis. We have made every effort to clarify this point in the revised manuscript.

The under-estimate of variance in the cross-validation tests

We acknowledge that proper estimation of variance is critical for robust and reliable cross-validation results. An important consideration in the context of the current analyses is that the permutation test used to assess statistical significance of support vector classification (SVC) and regression (SVR) results undergoes the same cross-validation procedure as that used in the original cross-validation. Therefore, the estimate of variance in the original cross-validation analysis is comparable to the null distribution from permutation. Importantly, this analytic approach accounts for any underestimate of variance in a modest sample size.

The likely challenges of reproducing the principle findings (in the absence of a independent test data set).

Reproducibility in neuroimaging research represents a major challenge for the study of pediatric clinical populations, such as children with ASD, whose data is considerably more difficult to acquire compared to typically-developing children (8). One consideration here is that while the sample size used in the current study (*N*=21 for both TD and ASD groups) is modest in comparison to recent task-based brain imaging studies of neurotypical adult populations and resting-state or structural studies in individuals with ASD, these types of studies do not face the same data collection challenges as task-based studies in clinical pediatric populations (8). Importantly, resting-state fMRI and structural MRI studies are unable to address specific questions related to social information processing in ASD, such as biologically-salient voice processing, which is critical for understanding the brain bases of social dysfunction in affected children. Furthermore, our sample size is larger than (9-12) or comparable to (13) other task-fMRI studies in children with ASD published since 2017.

Moreover, in task-fMRI studies published since 2017 that included >22 children with ASD, a much smaller number of runs (e.g., 2 runs of each condition (14); 4 runs (15); 1 run (16)) was provided to each participant compared to our study (7-10 runs with all stimulus conditions). This is an important consideration given that sample size is not the only determinant for the replicability of fMRI task data. While previous studies have shown that increasing the sample size can improve the replicability of results (17), an important consideration is that the replicability of task fMRI data is not solely contingent on a large sample size but also depends on the amount of individual-level sampling. A recent report examining this question showed that modest sample sizes, comparable to those described in our submitted manuscript, yield highly replicable results with only four runs of task data with a similar number of trials per run as our study (18). Moreover, replicability from smaller sample sizes using four runs of event-related task fMRI data exceeds the replicability of much larger sample sizes (*N* > 120) using only one run of block task fMRI data (18).

In the current study, we have used rigorous standards for inclusion that are, to the best of our knowledge, a first for autism and neurodevelopmental neuroimaging research. Specifically, we required that each child participant had *at least* 7 functional imaging runs of our event-related fMRI task that met our strict head movement criteria. This multi-run approach yields many more trials per vocal source condition (~150) than previous studies, thereby significantly enhancing power to detect effects within each child, and has only been used previously in visual neurosciences research in adults. To our knowledge, these rigorous within-subject criteria, which have been shown to be a critical factor for producing replicable task fMRI findings (18), are a first for autism and neurodevelopmental neuroimaging research.

The regression effect sizes are very strong, and there is possibly some colinearity between the test to identify the ROIs (the group contrast) and the subsequent regression (ASD severity), given that ASD typically presents as an extreme on the healthy development/social spectrum.

We believe that there is confusion regarding how we performed the ADOS covariate analysis within children with ASD. It seems that this confusion stems from the fact that reviewer #2 misunderstood the analysis and thought that the ADOS covariate analysis used ROIs that were identified from the TD vs. ASD group analysis. This was not the case. Rather, the TD vs. ASD group analysis and the ADOS covariate analysis were separate whole-brain analyses. By performing separate whole-brain analyses, we have avoided concerns related to collinearity in the ADOS covariate analysis. We have made every effort to clarify this point in the revised Materials and methods and Results sections. Finally, the strong effect sizes we found are likely the result of our use of *at least* 7 functional imaging runs in each participant.

Reviewer #1:

Overall, I very much enjoyed reading the paper: clearly written, timely, and novel. The use of recordings that are specific to each participant is an important contribution. I am not a neuroscientist so I am not qualified to evaluate the quality of the neuroimaging work. I will therefore focus my remarks on the Introduction, clinical / behavioural aspects of the Materials and methods, and Discussion. (and please pardon my naivety when I do comment on the neuroimaging part of the work).1) Behavioural tasks can provide mechanistic insightsI think the authors should tone down their claim about the limitations of behavioural studies to understand and tease apart different cognitive mechanisms. Many behavioural studies use ingenious paradigms to tease apart various mechanistic possibilities. Recently, the use of computational modeling methods applied to behavioural data has also been very fruitful in this respect (different model parameter reflect different mechanisms)

We thank the reviewer for raising this point. Our intention was to state that “behavioral studies are limited in their ability to provide insights into neural mechanisms underlying social information processing.” We have revised this section of the Introduction accordingly.

On the same topic, in the Introduction “…and obtaining valid behavioral measurements regarding individuals’ implicit judgments of subjective reward value of these stimuli can be problematic”: I was surprised by the strong statement. Antonia Hamilton has published many papers demonstrating the validity of behavioural tools to measure social reward responsiveness. I have also published several papers on that same topic (including one using signal detection theory, in PLOS One). In these papers, no abstract judgment is required. Rather, participants' behaviours are thought to reflect underlying social motivation or social reward responsiveness.

We thank the reviewer for highlighting this important point, and we agree with the reviewer that we should highlight this important line of research. An important consideration is that the behavioral work that has been done in this area by Drs. Chevalier (19, 20), Hamilton (21, 22), and others (23) has been in the visual domain, in which researchers use eye-tracking or pupillary responses as a tool for studying implicit reward. Studying implicit reward in the context of biologically-salient voice processing presents additional challenges given that, to our knowledge, validated behavioral methods for ascertaining whether children are directing their neural resources to a specific vocal source, thereby measuring auditory social reward responsiveness, have not yet been developed. Therefore, we argue that using brain imaging methods represents a critical approach in the context of implicitly rewarding voice processing. We have clarified these important points in the revised Introduction.

2) Why is the accuracy for mother's voice identification different in the ASD vs. TD group?I would like to know a bit more the difference in accuracy detection between the groups: why did the ASD group perform below the TD group? The authors point out that 5 children performed below chance in identifying their own mother's voice. Is there evidence that this indicates a true deficit or is poor performance linked to other factors (poor hearing? deteriorated listening skills in the scanner?). Where are these five children on the regression? Do they drive the effect?If some children did not recognize their mother's voice, it seems to be that they should be looked at differently: if the reward / memory / visual network is less activated in these children, is it because they do not find voices as rewarding / memorable or is it because they didn't recognize this familiar voice (but if they had, their brain would have reacted in the same way)?

Results and SI Results in the initial submission showed that differences in identification accuracy between TD and ASD groups for mother's voice identification were driven by 5 children with ASD who performed below chance on the “mother’s voice identification” task. The reviewer highlights an important point by suggesting that these 5 children might show a distinct behavioral or neural signature that may help explain why these children were unable to identify their mother’s voice in our behavioral task. While these children did not present with hearing impairments as noted by parents or neuropsychological assessors, who had performed extensive neuropsychological testing on these children prior to the fMRI scan and mother’s voice identification task, a plausible hypothesis is that the 5 children who were unable to identify their mother’s voice in the task would show greater social communication deficits, lower scores on measures of cognitive and language function, and/or reduced brain activation in response to unfamiliar or mother’s voice stimuli. To test this hypothesis, we have performed additional analyses to examine whether there are any identifying clinical, cognitive, or neural or characteristics regarding these 5 children with low identification accuracy.

We first investigated differences in social communication, cognitive, and language abilities between children with ASD with low (*N*=5) vs. high (*N*=16) mother’s voice identification accuracy. Examining the distribution of ADOS Social Communication revealed that the 5 children with low mother’s voice identification accuracy had a wide range of scores from 7-16 (please note that ADOS Social Communication is scored in a range between 0-20, with a score of 0 indicating no social deficit, a score of 7 indicating a more mild social communication deficit, and a score of 16 a more severe deficit). Group results for this measure are plotted below (left-most violin plot) and group comparisons between low (“Low ID” in green) and high (“High ID” in blue) mother’s voice identification groups using Wilcoxon rank sum tests were not significant for ADOS Social Communication (*P* = 0.83). We next examined whether there were any differences in standardized IQ scores (Wechsler Abbreviated Scale of Intelligence (24)) for children with low and high mother’s voice identification accuracy, which are plotted below (three right-most violin plots). Group comparisons between low and high mother’s voice identification groups using Wilcoxon rank sum tests were not significant for any of the IQ measures (*P* > 0.25 for all 3 measures, not corrected for multiple comparisons).

We next examined whether there were any differences for children with low vs. high mother’s voice identification accuracy in standardized measures of language abilities, including CTOPP Phonological Awareness (25) and CELF-4 Core Language, Receptive Language, and Expressive Language standard scores (26). Group comparison using Wilcoxon rank sum tests were not significant for any of the language measures (*P* > 0.05 for all 4 measures, not corrected for multiple comparisons), however there was a trend for reduced Core Language (*P* = 0.062) and Expressive Language abilities (*P* = 0.055) in the low (green) mother’s voice identification group.

**Author response image 2. respfig2:** 

We next examined neural response profiles for the 5 children with low vs. high mother’s voice identification accuracy by plotting ROI signal levels for the contrasts and regions identified in Figure 3A of the initial submission. First, results showed no group differences between children with low vs. high identification accuracy using Wilcoxon rank sum tests for any of the brain regions associated with the unfamiliar voices vs. non-social environmental sounds contrast (plotted below; *P* > 0.35 for all three regions, not corrected for multiple comparisons).

We next examined low vs. high identification accuracy using Wilcoxon rank sum tests for the brain regions associated with the mother’s voice vs. unfamiliar voices contrast (Figure 3B) and again found no group differences (plotted below; *P* > 0.45 for all seven regions, not corrected for multiple comparisons).

We next examined whether the 5 children with ASD with low mother’s voice identification accuracy showed distinct relationships between social communication and neural activation profiles compared to children with high mother’s voice identification accuracy, we replotted the Figure 3 regressions and demarcated children with low identification accuracy (plotted below with “X’s”). While this only provides a qualitative description of these relationships, results show a range of neural activation profiles for unfamiliar (panel A) and mother’s voice (panel B) processing in children with low mother’s voice identification accuracy. Importantly, the relationship between social communication and neural activation profiles did not appear to distinguish the children with low mother’s voice identification accuracy.

Finally, additional analyses, which were included in the initial submission, showed that removing the 5 children with low identification accuracy from the GLM and gPPI connectivity analyses did not change any of the reported effects (please see Appendix subsection entitled fMRI activation and connectivity profiles in children with ASD are not related to mother’s voice identification accuracy).

Together, results from clinical (i.e., ADOS Social Communication), cognitive, language, and neural activity measures showed that there are no distinguishing features for the children with poor mother’s voice identification accuracy. One possible explanation is that all children performed the mother's voice identification task immediately after the fMRI scan, which took approximately 2 to 2.5 hours to complete. The reason children performed this task after the fMRI scan rather than before it (i.e., at a neuropsychological testing visit prior to the scan) is that we did not want to expose the children to the fMRI stimuli prior to performing the fMRI task. Therefore, it seems plausible that these children may have had difficulty focusing on the mother’s voice identification task due to fatigue from the fMRI scan. We have included these additional analyses and information in the revised Appendix.

Another naive question on the same question. The authors report that "fMRI activation profiles in children with ASD were not related to mother's voice identification accuracy". So I was left wondering what these activations reflect (if they are not sensitive to the fact that 5 out of 21 children did not identify their mother, does it mean that these activations are picking up on something that is much more domain general than anything that might have to do with "mother's voice" specifically?)

This is an important question, and one that we have spent much time considering. If neural activations are “picking up on something that is much more domain general than anything that might have to do with ‘mother's voice’,” as indicated by the reviewer, we would hypothesize that a signature of domain general differences would be evident in supplementary analyses of behavioral and neural measures described in section 1.3 (above), including brain activation for [unfamiliar voices vs. non-social environmental sounds], which indexes general voice processing (27). However, results from supplementary analyses of social communication, cognitive, language, and neural measures of voice processing failed to provide evidence to suggest that the 5 children who could not accurately identify their mother’s voice are different from the other participants other than on the mother’s voice identification task. One additional domain-general possibility, which was discussed in section 1.3 (above), is that fatigue may have played a role in the 5 children who could not accurately identify their mother’s voice in the post scanning session. However, we do not have additional data to further probe this possibility.

Is there a way to statistically correct all analyses for accuracy levels?

To examine whether mother’s voice identification accuracy affected results from ADOS covariate analyses in children with ASD (Figure 3), we performed additional regression analyses in which ADOS social communication values were the dependent variable and predictors included mother’s voice identification accuracy and betas from ROIs identified in the [unfamiliar female voice minus environmental sounds] contrast (i.e., Figure 3A) or [mother’s voice minus unfamiliar voices] contrast (i.e., Figure 3B). Separate regression models were computed for each ROI in each contrast. Results showed that all ROI signal levels reported in Figure 3 were significant predictors of ADOS social communication scores after regressing out mother’s voice identification accuracy (P ≤ 0.005 for all ROIs). We have added these results to the revised Appendix subsection entitled fMRI activation and connectivity profiles in children with ASD are not related to mother’s voice identification accuracy.

The authors should report whether identification accuracy is related to ADOS SC scores.

We thank the reviewer for this suggestion and correlation analysis indicates that mother’s voice identification accuracy is not related to ADOS social communication scores (*R* = 0.13, *P* = 0.59). We have included this result in the revised Appendix.

3) ADOS score useADOS scores are not meant to be used as a continuous severity scores unless they are transformed (Gotham, K., Pickles, A., & Lord, C. (2009). Standardizing ADOS scores for a measure of severity in autism spectrum disorders. Journal of autism and developmental disorders, 39(5), 693-705.). I do not know how this logic applies to using the social affect score only but it should be checked / discussed.

We thank the reviewer for this astute point. Standardization of ADOS scores (28) and subscores (29) is performed to enable comparisons across ADOS modules. Given that all of our participants were administered module 3 of the ADOS, which is stated in the Participants subsection of the Materials and methods, standardization of ADOS scores here is unnecessary.

4) Are there a priori criteria for exclusion based on motion?Materials and methods subsections “Participants” and “Movement criteria for inclusion in fMRI analysis”: Is there an a priori threshold to exclude participants based on motion? Or published guidelines? Can the authors specify the exclusion decision criteria? Is there a group difference in average motion?

Our study incorporated stringent a priori criteria for exclusion based on head motion during all fMRI runs, which is consistent with our previous work (30) and is described in the Materials and methods subsection entitled Movement criteria for inclusion in fMRI analysis. This sections states:

“For inclusion in the fMRI analysis, we required that each functional run had a maximum scan-to-scan movement of < 6 mm and no more than 15% of volumes were corrected in the de-spiking procedure. Moreover, we required that all individual subject data included in the analysis consisted of at least seven functional runs that met our criteria for scan-to-scan movement and percentage of volumes corrected; subjects who had fewer than seven functional runs that met our movement criteria were not included in the data analysis. All 42 participants included in the analysis had at least 7 functional runs that met our movement criteria, and the total number of runs included for TD and ASD groups were similar (TD = 192 runs; ASD = 188 runs).”

The second to last line in Table 1: Demographic and IQ Measures shows descriptive statistics for head motion in the TD and ASD groups as well as results from a group comparison using a 2-sample t-test, which was not significant (P = 0.36). We have further clarified this point in the revisedParticipants subsection of the revised Materials and methods.

5) Sample size concernsIn a behavioural study, a sample size of 21 is now considered too small. I realise that the change in standard is recent (and I have published many papers using small sample sizes myself). However, I do think that we need to accelerate change, especially for conditions that are notoriously heterogenous, such as ASD. And especially when one is interested in explaining interindividual differences (by using correlations).

We thank the reviewer for this comment and their concern for the ability of our study to address heterogeneity in ASD. Here we have addressed this challenge with a comprehensive approach which both: (a) constrains heterogeneity in ASD to a critical diagnostic symptom domain, social communication abilities, and (b) explores heterogeneity within this symptom domain (i.e., “explaining interindividual differences” as mentioned by the reviewer) by identifying neural features that covary as a function of social communication abilities. The rationale for constraining heterogeneity in ASD to a symptom domain is that all individuals with ASD necessarily have pronounced impairments in diagnostic domains, and these domains have considerably less heterogeneity in individuals with ASD relative to other behavioral and cognitive domains. For example, in the social communication domain, which we explore in the current study, all individuals with ASD show pronounced deficits that are categorized by impaired communication and/or reciprocal social interactions (1). In contrast, please consider language function and IQ in children with ASD. Language function and IQ in these individuals varies widely, from very high levels, which are commensurate with neurotypical individuals, through very low levels of language and cognitive function (1-3). We argue that constraining analyses to an ASD diagnostic domain is an important approach for keeping a focus on critical areas that are central to the core deficits in the disorder, thereby providing important insight to clinicians, researchers, parents, and educators regarding the neurobiological bases of these core deficits.

The rationale for exploring heterogeneity within the social communication symptom domain is that high-functioning children with ASD, such as those included in our sample, often show a range of social communication abilities, from mild/moderate deficits to more severe deficits. For example, the Autism Diagnostic Observation Schedule (ADOS-2) is the gold-standard diagnostic instrument for ASD (4), and social communication subscores on the ADOS range from 0 to 22, with a 0 indicating no social communication deficit, a 7 indicating a more mild deficit, and a 22 indicating the most severe deficit. The high-functioning children included in our sample had a range of social communication abilities as measured with the ADOS between 7 and 16. Importantly, understanding the link between social communication symptom severity and social brain processing is a critical question for understanding the neurobiological basis of autism, and is a question that has not been explored in previous studies of human voice processing in children with ASD. The reason this is an important question is that the severity of social communication deficits can play a crucial role in autism models. For example, the social motivation theory of ASD posits that impairments in representing the reward value of human vocal sounds impedes individuals with ASD from engaging with these stimuli and contributes to social interaction difficulties (5, 6). Given the causal link proposed in this model, a key prediction of the social motivation theory is that children with more severe deficits associated with social reward will have greater social communication deficits compared to those children with less severe social reward deficits. Therefore, by examining heterogeneity within the social communication symptom domain, we are able to test an important prediction of an influential ASD model and understand the neural features that may contribute to this heterogeneity.

In our study, we have employed this approach and constrained heterogeneity in children with ASD by focusing on social communication abilities. We then examine heterogeneity within the social communication symptom domain and show that social communication abilities explain significant variance in activity and connectivity of social reward and salience brain regions during human voice processing.

Finally, it is important to note that our inclusion criteria required each participant to have at least 7 functional imaging runs of our event-related fMRI task that met our strict head movement criteria. Requiring a large number of functional runs for each participant is an important approach for increasing statistical power, an issue further elaborated below. This approach has been used primarily in basic human vision experiments, which often use small samples of 3-7 participants with intense scanning in each participant (31-33), and, to our knowledge, is a first for neuroimaging studies in autism, and in children.

Reviewers who specialise in neuroimaging methods may want to comment on this specific point but the paper would definitely be stronger is the sample size exceeded (or at least matched) the current average in cognitive neuroscience field (ie 30, see Poldrack et al. 2015, Figure 1).

Reproducibility in neuroimaging research represents a major challenge for the study of pediatric clinical populations, such as children with ASD, whose data is considerably more difficult to acquire compared to typically-developing children (8). One consideration here is that while the sample size used in the current study (*N*=21 for both TD and ASD groups) is modest in comparison to recent task-based brain imaging studies of neurotypical adult populations and resting-state or structural studies in individuals with ASD, these types of studies do not face the same data collection challenges as task-based studies in clinical pediatric populations (8). Importantly, resting-state fMRI and structural MRI studies are unable to address specific questions related to social information processing in ASD, such as biologically-salient voice processing, which is critical for understanding the brain bases of social dysfunction in affected children. Furthermore, our sample size is larger than (9-12) or comparable to (13) other task-fMRI studies in children with ASD published since 2017. Moreover, in task-fMRI studies published since 2017 that included >22 children with ASD, a much smaller number of runs (e.g., 2 runs of each condition (14); 4 runs (15); 1 run (16)) was provided to each participant compared to our study (7-10 runs with all stimulus conditions). This is an important consideration given that sample size is not the only determinant for the replicability of fMRI task data. While previous studies have shown that increasing the sample size can improve the replicability of results (17), an important consideration is that the replicability of task fMRI data is not solely contingent on a large sample size but also depends on the amount of individual-level sampling. A recent report examining this question showed that modest sample sizes, comparable to those described in our submitted manuscript, yield highly replicable results with only four runs of task data with a similar number of trials per run as our study (18). Moreover, replicability from smaller sample sizes using four runs of event-related task fMRI data exceeds the replicability of much larger sample sizes (*N* > 120) using only one run of block task fMRI data (18).

In the current study, we have used rigorous standards for inclusion that are, to the best of our knowledge, a first for autism and neurodevelopmental neuroimaging research. Specifically, we required that each child participant had at least 7 functional imaging runs of our event-related fMRI task (4 min each) that met our strict head movement criteria. This multi-run approach yields many more trials per condition (~150) than previous studies, thereby significantly enhancing power to detect effects within each child, and has only been used previously in visual neurosciences research in adults. To our knowledge, these rigorous within-subject criteria, which have been shown to be a critical factor for producing replicable task fMRI findings (18), are a first for autism and neurodevelopmental neuroimaging research.

Alternatively, the authors should report observed power. This is I think most important for the regression part of the paper. If power is too low, I would recommend moving this part of the paper to the SM and rewriting the main text accordingly.

To provide a better estimate of effect size, we used the originally computed *t*-scores from the whole-brain GLM analysis. Instead of examining the peak, we averaged the *t*-scores in each cluster to compute effect sizes. To estimate effect sizes for the TD vs. ASD group comparisons (i.e., regions identified in Figure 2), *t*-scores from the whole-brain TD vs. ASD group GLM analysis were averaged within each cluster identified in the GLM results. Effect sizes were then computed as Cohen’s *d* = *t*-scores/(sqrt(*N/2*)), where *t* is the mean t-score within a cluster *N* is the sample size.

To provide estimates of effect sizes within regions identified in the ASD Social Communication covariate analysis (i.e., Figure 3), *t*-scores from the whole-brain covariate analysis were averaged within each cluster identified in the results. Effect sizes were then computed as Cohen’s *f* according to *f* = *t*-scores/(sqrt(*N*)), where *t* is the mean *t*-score within a cluster and N is the ASD sample size: These effect sizes are now reported in the revised manuscript. We report an overall effect size of 0.68 averaged across all clusters identified in the TD vs. ASD group analysis (Figure 2) and an overall effect size of 0.76 averaged across all clusters identified in the ASD Social Communication Covariate analysis (Figure 3).

Reviewer #2:

Impaired voice processing in.… children with autism" by Abrams et al.This paper reports the uni- and multivariate analysis of fMRI acquired from children with ASD while listening to their mothers' or strangers' voice. The authors find quite striking correlations between social disability rating scales and both brain activity and functional connectivity.The experimental stimuli, writing, analysis and cohort characterization are all of a very high standard. There are some limitations in the actual design that limit the nature of the inferences drawn. In a revised ms, these should be addressed through a subtle reframing and possibly additional discussion points.1) The task stimuli are very well controlled but the task is essentially a passive listening task, with a low level vigilance task designed simply to keep the participants engaged with the stimuli. Therefore, in the absence of an explicit reward, the authors should be cautious of falling foul of reverse inference (Poldrack, 2006) when framing the findings in terms of "reward circuits". I guess listening to a voice, particularly a mother's voice, is implicitly rewarding, but caution should be taken in drawing to direct an inference in regards to dysfunctional reward processes (see Discussion section).

We thank the reviewer for this comment and acknowledge that, as with all naturalistic and biologically salient stimuli, we cannot know for certain whether aberrant activation and connectivity patterns measured in nucleus accumbens (NAc) and ventromedial prefrontal cortex (vmPFC) in children with ASD reflect reward processing in these regions. However, previous empirical evidence and theory, which we have highlighted in our manuscript, provide a strong theoretical foundation for considering vocal stimuli in the context of reward, even in the absence of an explicit reward task. First, there is a sizable behavior literature that shows the implicitly rewarding nature of the human voice, including mother’s voice, in neurotypical children (34-38). Second, behavioral evidence shows that children with ASD often fail to be attracted to human vocal sounds, even when they are able to engage with other sounds in their environment, which suggests that they may not find these sounds rewarding (39, 40). Third, an influential theory posits that social reward processing, such as weak reward attribution to vocal communication, may substantially contribute to pronounced social deficits in children with ASD (5, 6). Given these converging results and theory, we believe that considering reward in the context of diminished activity and connectivity in response to vocal sounds in brain regions that are closely associated with reward processing (i.e., NAc and vmPFC) in children ASD is an important hypothesis. Importantly, we have made every effort to temper statements to avoid issues with reverse inference in the revised manuscript. Specifically, we have used “results suggest…” or “results support/are consistent with the hypothesis that…” in all instances throughout the Abstract, Introduction, Results, and Discussion in which we discuss “reward” in the context of activity or connectivity associated with NAc or vmPFC.

Indeed, the whole positioning of functional neuroimaging studies as being more informative than behavioural tests should be mindful of the limitations of inferring disrupted functional circuits in disorders unless you are actually probing that function with an appropriate task.

We apologize for any confusion here. Our goal for these statements was to highlight neuroimaging research as an additional tool to test important hypotheses regarding voice and reward processing in children with ASD. We have clarified these statements in the revised manuscript.

2) Similarly, there is no manipulation of attention and therefore no means of telling whether the differences in activation during voice perception in ASD is simply a lack of due interest in, and attention to voices preceding (or consequent) to changes in reward-based learning.

While parametrically manipulating attention to human voices in children with ASD is an important question, unfortunately this was beyond the scope of the current work. Importantly, we hypothesize that a lack of interest in human voice processing is a fundamental prediction of the social motivation theory (5): humans engage and pay attention to rewarding stimuli in their environment, and a consistent lack of attention to a category of stimuli strongly suggests that these stimuli may not be rewarding to an individual (5). It is hoped that future studies will test this prediction and examine the relative contributions of attention and reward for human voices in children with ASD.

3) Again the authors draw a very direct line between mother's vs stranger's voice and social reasoning/communication (e.g. Abstract). But maternal voice is more than a simple cue but incorporates many other processes, including basic parental attachment and dyadic reciprocation.

We thank the reviewer for requesting clarification here. The relationship we had intended to highlight here was that of the link between social communication abilities in children with ASD and brain activation in social and reward brain areas during voice processing. We have clarified this important point in the revised manuscript.

4) What are the core defining disturbances underlying the diagnosis of ASD in this study? The information in paragraph two of subsection “Participants” simply refers to an algorithm.

Autism spectrum disorder is characterized by pronounced social communication deficits, particularly in the areas of social-emotional reciprocity and verbal and non-verbal communication, and repetitive and restricted behaviors (RRB) and interests (1). As stated in the Participants subsection of the Materials and methods, the children in the ASD sample are considered “high-functioning” and have fluent language skills, normal IQ, and above-average reading skills. Nevertheless, these children are generally characterized as having moderate-to-severe communication impairments, especially in the area of reciprocal conversation (1). We have included additional information regarding the defining characteristics of ASD in the revised Participants section.

While this might be sufficient for reproducibility, it is inadequate here for two reasons: First, as eLife is a general (not a clinical) journal, more depth and context is required for the broader readership. Second, if the diagnosis relies heavily on social deficits (which I suspect it does), is it then surprising that the strength of between group differences covaries so markedly within the ASD with social deficit scores. If so, is there an implicit circularity between the identification of these regions (Figure 2) and the very strong correlations against ADOS scores? If not, what other effects may be driving these? Such strong correlations are bound to draw attention and I think the authors should therefore pay careful attention to this issue.

We believe that there is confusion regarding how we performed the ADOS covariate analysis within children with ASD. It seems that this confusion stems from the fact that the reviewer misunderstood the analyses and thought that the ADOS covariate analysis used ROIs that were identified from the TD vs. ASD group analysis. This was not the case. Rather, the TD vs. ASD group analysis and the ADOS covariate analysis were separate whole-brain analyses. The use of separate whole-brain analyses in this context avoids circularity mentioned by the reviewer. We have made every effort to clarify this point in the revised Materials and methods and Results sections.

5) This is a modest sample size for the use of machine learning, particularly when using a high dimensional (functional connectivity) feature space. N-fold cross-validation does always not provide strong control here (Varoquaux, 2017): Do the authors undertake a feature reduction step, such as a LASSO?

We thank the reviewer for highlighting this important point. To examine the robustness of SVC and SVR results reported in the Results, a confirmatory analysis was performed using GLMnet (http://www-stat.stanford.edu/~tibs/glmnet-matlab), a logistic regression classifier that includes regularization and includes a feature reduction step. Results from GLMnet were similar to those reported for SVC and SVR results, and were reported in Results section of the initial submission of this manuscript.

Is there some logical circularity between identifying features with a group contrast, then using these same features in a between group classifier?

We thank the reviewer for inquiring about this point. As we stated in Materials and methods section of the initial submission:

“The rationale for the use of an a priori network is it is an established method of network identification that preempts task and sample-related biases in region-of-interest (ROI) selection. This approach therefore allows for a more generalizable set of results compared to a network defined based on nodes identified using the current sample of children and task conditions.”

We believe that this is not a circular approach for two reasons: (1) the group contrast used to identify ROIs for the functional connectivity analysis was from a previous study (41) in an independent sample of children with ASD relative to the participants in the current study, and (2) the brain imaging approach and analysis employed in that previous study was intrinsic functional connectivity using resting-state data and seed-based analyses, which provides complementary information regarding brain network organization relative to the task-based data and gPPI analysis used in the current study. The importance of this approach was highlighted in the subsection entitled A voice-related brain network approach for understanding social information processing in autism in the Discussion section of the initial submission. We would like to bring special attention to the final sentence in this paragraph (quoted below) which highlights the fact that the networks approach used in the current study bridges a critical gap between findings from intrinsic connectivity analyses (41) and task-based social information processing that is fundamental to social communication deficits in children with ASD.

“A central assumption of [the intrinsic functional connectivity] approach is that aberrant task-evoked circuit function is associated with clinical symptoms and behavior, however empirical studies examining these associations have been lacking from the ASD literature. Our study addresses this gap by probing task-evoked function within a network defined a priori from a previous study of intrinsic connectivity of voice-selective networks in an independent group of children with ASD. We show that voice-related network function during the processing of a clinically and biologically meaningful social stimulus predicts both ASD group membership as well as social communication abilities in these children. Findings bridge a critical gap between the integrity of the intrinsic architecture of the voice-processing network in children with ASD and network signatures of aberrant social information processing in these individuals.”

Also, I would avoid using the term "prediction" for contemporaneous variables, even when cross-validation is undertaken.

Consistent with many papers in the fMRI literature (42-45), the use of prediction to describe cross-validated results is a widely used convention and therefore we would prefer to use this nomenclature in our study.

6) I don't see any model-based analysis of neuronal interactions and hence don't think the authors are examining effective connectivity. gPPI is a purely linear model of statistical dependences and their moderation and hence falls into the class of functional connectivity.

We appreciate this suggestion and have removed all instances of “effective connectivity” in the revised manuscript.

Personally, I would prefer to have seen the author use a more dynamic, model-driven method of effective connectivity, using something like DCM to provide a deeper mechanistic insight into the changes in activation and information flow

While we share the reviewer’s interest in providing a deeper mechanistic insight into voice processing in children with ASD, we had significant concern regarding the implementation of DCM in the context of the relatively long TR (3.576 seconds) used in data collection. The reason for this long TR is that it allowed the auditory stimuli to be presented in silent periods between volume acquisitions. Moreover, serious concerns have been raised in the literature regarding DCM (46) especially when estimating causal influences with a large set of nodes as we did in the present study. The gPPI models used in our study are not faced with estimability issues.

I also am not sure I buy into the choice of ROI's that do not show the group effects.

The reason that ROIs were not selected based on a group effect is that this approach could be considered circular, and our goal was to provide a more generalizable set of results compared to a network defined based on nodes identified using the current sample of children and task conditions. The use of an a priori network is it is an established method of network identification that preempts task and sample-related biases in region-of-interest (ROI) selection (47-50).

However, given the classification success, the authors' approach seems entirely reasonable. However, please do specify the nature of the gPPI model in the text and supplementary material (what are the nodes, inputs and modulators; is it possible to represent this graphically?).

We examined functional connectivity between ROIs using the generalized psychophysiological interaction (gPPI) model (51), with the goal of identifying connectivity between ROIs in response to each task condition as well differences between task conditions (mother’s voice, other voice, environmental sounds). We used SPM gPPI toolbox for this analysis. gPPI is more sensitive than standard PPI to task context-dependent differences in connectivity (51). Unlike dynamical causal modeling (DCM), gPPI does not use a temporal precedence model (x(t+ 1) ~ x(t)) and therefore makes no claims of causality. The gPPI model is summarized in Equation 1 below:

ROItarget~convdeconvROIseed*taskwaveform+ROIseed+constant(1)

Briefly, in each participant, the regional timeseries from a seed ROI is deconvolved to uncover quasi-neuronal activity and then multiplied with the task design waveform for each task condition to form condition-specific gPPI interaction terms. These interaction terms are then convolved with the hemodynamic response function (HRF) to form gPPI regressors for each task condition. The final step is a standard general linear model predicting target ROI response after regressing out any direct effects of the activity in the seed ROI. In the equation above, ROItarget and ROIseed are the time series in the two brain regions, and taskwaveformcontains three columns corresponding to each task condition. We have included this description in the revised *Functional Connectivity Analysis* subsection of the Materials and methods.

7) Do the manipulations to the vocal signals (subsection “Stimulus post-processing”) possibly warp the sound of the speech? If so, could this influence the ASD responses (ASD being perhaps more tuned to low level features of stimulus inputs).

Manipulations to the vocal signals during stimulus preparation were minimal and were performed on all mother’s and unfamiliar voice and environmental sound stimuli included in the study. We hypothesize that significantly warped vocal samples would have resulted in reduced mother’s voice identification accuracy is at least one of the TD children, however results showed that all TD children performed above chance on this task, with 20 of 21 TD children revealing >90% identification accuracy on this task (mean mother’s voice identification accuracy in TD children was 98%; see Table 1). Moreover, while there are mixed reports of increased auditory discrimination in individuals with ASD (52-55), with studies showing a relatively small subgroup (~20%) of individuals with ASD with enhanced auditory perceptual abilities (i.e., “more tuned to low level features of stimulus inputs” as suggested by the reviewer), it is not immediately clear how these enhanced auditory perceptual abilities would diminish one’s ability to discriminate mother’s voice. An arguably more likely possibility is that established deficits in children with ASD associated with phonological abilities (2, 56-58), which involve the processing of the sound structure of language, would have been linked to reduced mother’s voice discrimination accuracy, however results reported above (see reviewer 1) and now included in the Appendix (Appendix 1—figure 1), failed to show a difference in phonological abilities in children with low and high mother’s voice identification accuracy.

Minor Comments:1) Introduction, second sentence: "affected" -> "ASD"

Done

2) Figure 1B caption: Please add stats

Done

3) Discussion first paragraph: "These findings.…"; Third paragraph: "Our findings.…"

Done

4) Discussion paragraph three: ?"predicts"?

As we stated previously, the use of the word “prediction” to describe cross-validated results is a widely used convention (42-45), and therefore we would prefer to continue to use this nomenclature in our study.

5) Subsection “Participants”: please add more details regarding the diagnosis; e.g. "In essence, these ASD.…" (see point 4 above)

Done.

6) Table 1: What are the "typical" values of ADOS-social and ADI-A social"? Are these very impaired children (also see point 4 above).

We have added this information to the *Participants* subsection of the revised Materials and methods.

7) Materials and methods: Were there any between group differences in head motion parameters?

No, there were no between-group differences in head motion parameters. Mean and standard deviation for maximum head motion for TD and ASD groups are included in Table 1, and we have further clarified this point in the revised*Participants* subsection of the revised Materials and methods.

8) Subsection “Effective Connectivity Analysis”: I'm not sure what "preempts.… biases" means here – the authors are not modelling effects in the present data, which I think is a shame (see point 6 above)

Network identification in brain imaging studies presents several challenges. One important consideration is that selecting ROIs based on a GLM contrast from a sample of participants may be considered circular when those ROIs will then be used in a subsequent functional connectivity analysis on that same contrast and sample of participants. Therefore, by using ROIs from a previous study of the intrinsic architecture of voice-selective cortex in a separate sample of children with ASD (41), we have preempted biases associated with both task contrast and participant sample that would have emerged had we used task-based GLM results from the current sample to generate ROIs. Finally, we have provided an explanation for why we have not performed additional causal analyses in section 2.11 above.

Reviewer #3:

In this study, Abrams and colleagues examined voice processing in children (average age 10 years old) with and without autism. The specifically were interested in reward and salience circuitry and relate the study back to predictions made by the social motivation theory of autism. The authors applied group-level analyses, brain-behavior correlation analyses, as well as PPI connectivity analyses and applied multivariate classifier and regression analyses applied to the connectivity data. There are several issues which the authors may want to address.1) Small sample size. The sample size for the study was n=21 per group. This sample size is likely not large enough to cover substantial heterogeneity that exists across the population of individuals with autism diagnosis.

We thank the reviewer for this comment and their concern for the ability of our study to address heterogeneity in ASD. As discussed in response to similar reviewer comments, here we have addressed this challenge with a comprehensive approach which both: (a) constrains heterogeneity in ASD to a critical diagnostic symptom domain, social communication abilities, and (b) explores heterogeneity within this symptom domain (i.e., “explaining interindividual differences” as mentioned by the reviewer) by identifying neural features that covary as a function of social communication abilities. The rationale for constraining heterogeneity in ASD to a symptom domain is that all individuals with ASD necessarily have pronounced impairments in diagnostic domains, and these domains have considerably less heterogeneity in individuals with ASD relative to other behavioral and cognitive domains. For example, in the social communication domain, which we explore in the current study, all individuals with ASD show pronounced deficits that are categorized by impaired communication and/or reciprocal social interactions (1). In contrast, please consider language function and IQ in children with ASD. Language function and IQ in these individuals varies widely, from very high levels, which are commensurate with neurotypical individuals, through very low levels of language and cognitive function (1-3). We argue that constraining analyses to an ASD diagnostic domain is an important approach for keeping a focus on critical areas that are central to the core deficits in the disorder, thereby providing important insight to clinicians, researchers, parents, and educators regarding the neurobiological bases of these core deficits.

The rationale for exploring heterogeneity within the social communication symptom domain is that high-functioning children with ASD, such as those included in our sample, often show a range of social communication abilities, from mild/moderate deficits to more severe deficits. For example, the Autism Diagnostic Observation Schedule (ADOS-2) is the gold-standard diagnostic instrument for ASD (4), and social communication subscores on the ADOS range from 0 to 22, with a 0 indicating no social communication deficit, a 7 indicating a more mild deficit, and a 22 indicating the most severe deficit. The high-functioning children included in our sample had a range of social communication abilities as measured with the ADOS between 7 and 16. Importantly, understanding the link between social communication symptom severity and social brain processing is a critical question for understanding the neurobiological basis of autism, and is a question that has not been explored in previous studies of human voice processing in children with ASD. The reason this is an important question is that the severity of social communication deficits can play a crucial role in autism models. For example, the social motivation theory of ASD posits that impairments in representing the reward value of human vocal sounds impedes individuals with ASD from engaging with these stimuli and contributes to social interaction difficulties (5, 6). Given the causal link proposed in this model, a key prediction of the social motivation theory is that children with more severe deficits associated with social reward will have greater social communication deficits compared to those children with less severe social reward deficits. Therefore, by examining heterogeneity within the social communication symptom domain, we are able to test an important prediction of an influential ASD model and understand the neural features that may contribute to this heterogeneity.

In our study, we have employed this approach and constrained heterogeneity in children with ASD by focusing on social communication abilities. We then examine heterogeneity within the social communication symptom domain and show that social communication abilities explain significant variance in activity and connectivity of social reward and salience brain regions during human voice processing.

Thus, questions about generalizability arise. Can future studies replicate these findings?

Reproducibility in neuroimaging research represents a major challenge for the study of pediatric clinical populations, such as children with ASD, whose data is considerably more difficult to acquire compared to typically-developing children (8). An important consideration is that sample size is not the only determinant for the replicability of fMRI task data. While previous studies have shown that increasing the sample size can improve the replicability of results (17), an important consideration is that the replicability of task fMRI data is not solely contingent on a large sample size but also depends on the amount of individual-level sampling. A recent report examining this question showed that modest sample sizes, comparable to those described in our submitted manuscript, yield highly replicable results with only four runs of task data (18). Moreover, replicability from smaller sample sizes using four runs of event-related task fMRI data exceeds the replicability of much larger sample sizes (*N* > 120) using only one run of block task fMRI data (18).

In the current study, we have used rigorous standards for inclusion that are, to the best of our knowledge, a first for autism and neurodevelopmental neuroimaging research. Specifically, we required that each child participant had at least 7 functional imaging runs of our event-related fMRI task (4 min each) that met our strict head movement criteria. This multi-run approach yields many more trials per condition (~150) than previous studies, thereby significantly enhancing power to detect effects within each child, and has only been used previously in visual neurosciences research in adults. To our knowledge, these rigorous within-subject criteria, which have been shown to be a critical factor for producing replicable task fMRI findings (18), are a first for autism and neurodevelopmental neuroimaging research.

Small sample size also means lower statistical power for identifying more subtle effects, and this is especially important for the context of whole-brain between-group or brain-behavior correlation analysis which the authors have solely relied on for the activation and clinical correlation analyses (see Cremers, Wager, & Yarkoni, 2017, PLoS One).

We thank the reviewer for drawing our attention to these important concepts. From one perspective, identifying more subtle effects comes with its own challenges: weak effects are often viewed with caution irrespective of statistical power. As stated by Reddan, Lindquist, & Wager, (2017, JAMA Psychiatry), “small effects can reach statistical significance given a large enough sample, even if they are unlikely to be of practical importance or replicable across diverse samples.”

While we agree that a larger sample size would have been preferred, we argue that the rigorous within-subject criteria implemented for the current study, which is described in detail in response to the points raised in 3.2 above and is a first for neuroimaging studies of children with ASD, bolsters the ability for this study to identify more subtle GLM effects (18). It should also be noted that we also identified significant between-group (Figure 4) and brain-behavior relationships (Figure 5) using an a priori brain network identified from an independent sample of children with ASD (41).

For multivariate classifiers and regressions, these too produce inflated and over-optimistic levels of predictions with smaller sample size (e.g., Woo et al., 2017, Nature Neuroscience).

We again thank the reviewer for this comment regarding sample size, and have considered the important report by Woo et al. (2017) in the context of our study. While we agree that a larger sample size would have been preferable, we note that the current study avoided analysis procedures identified by Woo et al. that are performed across the dataset before training and testing data (e.g., denoising, scaling, component analyses, and feature selection) and that can create “dependence and optimistic biases in [cross-validated] accuracy”. Crucially, as discussed previously, we required that each child participant had at least 7 functional imaging runs of our event-related fMRI task (4 min each) that met our strict head movement criteria. The acquisition of high quality brain imaging data, including at least 7 functional runs from each child, is extremely difficult in the context of pediatric clinical populations (8) and is unprecedented in studies of autism.

2) Given the small sample sizes, but relatively strong and justified anatomical hypotheses, why not run ROI analyses instead of whole-brain analyses? Statistical power would likely be increased for ROI analyses, and one can cite more unbiased estimates of effect size. Whole-brain analyses can show us is where the likely effects might be, and this is helpful when we don't have strong anatomical hypotheses. But here the authors do have strong anatomical hypotheses, yet they choose an analysis approach that is not congruent with that and penalizes them in terms of statistical power and doesn't allow for estimation of unbiased effect sizes.

We thank the reviewer for this remark. We did in fact perform ROI analyses on the GLM results using ROIs from our previous intrinsic functional connectivity paper in children with ASD (41), however several issues emerged when we used this approach. First, intrinsic connectivity results from this previous study identified 16 ROIs, and including all of these ROIs would have required FDR correction, which would have reduced the ability to detect subtle effects. Furthermore, had we limited the number of ROIs included in the analysis to reduce the multiple comparisons issue, it may have appeared that we were selecting and reducing the ROIs after results are known (i.e., SHARKing (59)). Finally, results showed that there was not exact overlap between ROIs from our previous intrinsic functional connectivity study and GLM effects identified in the current study for unfamiliar and mother’s voice contrasts, which resulted in GLM activity in reward and affective processing regions (i.e., NAc, vmPFC, anterior insula, anterior cingulate cortex) that did not yield significant GLM results in the *a priori* ROIs. Consequently, we did not report ROI-based GLM results in the initial submission of the manuscript.

While the use of ROIs based on anatomical hypotheses would have been justified for GLM analyses, we do not believe that the use of whole-brain analysis in the context of the current study presents a methodological weakness. Consistent with the reviewer’s statement, the use of whole-brain analysis penalized our ability to identify effects, and despite this penalty, GLM results identified effects in reward and salience processing regions in response to vocal sounds in children with ASD.

What is missing from the paper is an estimate of how big the effects are likely to be, as this is what we should ideally care about (Reddan, Lindquist, & Wager, 2017, JAMA Psychiatry). Future studies that may try to replicate this study will need to know what the effect sizes are likely to be. Meta-analyses ideally need unbiased estimates of effect size. However, all we have to go off of here are the authors figures showing whole-brain maps, that likely just tell us where some of the largest effects may likely be.3) Reported effect sizes in Figure 3 (e.g., Pearson's r) are likely inflated given small sample sizes and also due to the fact that it appears that the reported r values in the figure are likely taken from the peak voxel.

The plots in Figure 3 were meant to aid visualization of regional brain responses, and were not intended to reflect effect size. As noted in the excellent report by Reddan, Lindquist, & Wager, effects sizes in brain imaging studies are prone to numerous biases including the number of tests performed and number of brain regions/voxels examined, and the specific brain regions selected. In general, there is no good solution to this problem. A contributing factor is the more stringent GLM activation thresholds that are published in more recent fMRI papers: effect sizes increase when higher voxel-wise *t*-scores are used to compute them – there is no good solution to this problem. To provide a better estimate of effect size, we used the originally computed *t*-scores from the whole-brain GLM analysis. Instead of examining the peak, we averaged the *t*-scores in each cluster and computed the effect size = *t*-scores/(sqrt(*N*)), where *N* is the sample size. In the revised manuscript, these effect sizes have replaced the *R* and *P* values previously listed in Figure 3 scatterplots, and are also provided in Appendix 1—tables 1 and 2. To provide additional guidance for future studies that seek to replicate our findings, we report an overall effect size of 0.68 averaged across all brain regions examined in the TD vs. ASD group analysis (Figure 2) and an overall effect size of 0.76 averaged across all brain regions examined in the ASD Social Communication Covariate analysis (Figure 3).

4) Scatterplots in Figure 3 show inverted y-axes so that higher numbers on at the bottom and lower numbers are at the top. The reported correlations are negative, and yet the scatterplot shows what looks like a positive correlation. All this confusion is due to the inverted y-axes. The authors should correct the plots to avoid this confusion.

We inverted the y-axes in the initial submission since greater values of ADOS Social Communication scores are associated with more severe deficits, and often readers prefer to see reduced abilities at the bottom of the y-axis rather than at the top. If the reviewer still feels that we should make the change, we would be happy to.

5) The authors heavily rely on reverse inference to relate their findings back to the social motivation theory. However, if their manipulations were powerful enough to create a distinction between a stimulus that was heavily socially rewarding (e.g., mother's voice) versus another that is not (e.g., unfamiliar voice), then shouldn't there be some kind of difference in the main activation analysis in reward-related areas (i.e. Figure 2B)? In other words the main contrast of interest that might have been most relevant to the social motivation theory produces no group differences in activation in areas like the ventral striatum. Because this contrast doesn't really pan out the way the theory predicts, doesn't this cast doubt on the social motivation theory, or couldn't it be that some of the contrasts in this study can be better explained by some other kind of reverse inference than the social motivation theory?

We thank the reviewer for this comment. Our interpretation of the results is that there was a high degree of variance within the ASD group with regards to neural responses to vocal stimuli, and that variance within the ASD group was comparable to (or exceeded) the between-group variance, prohibiting significant between-group differences. This interpretation is supported by results in TD children (Appendix 1—figure 4A) and the scatterplots in Figure 3 showing the relationship between ADOS Social scores and neural activation profiles in children with ASD. While TD children, who do not have social deficits, showed robust responses in NAc and vmPFC in response to vocal stimuli (Appendix 1—figure 4A), activity in these regions was not evident in the ASD group (Appendix 1—figure 4B) until ADOS Social scores were included as a covariate in the analysis (Main Figure 3). Specifically, these latter results showed that the greater the social function in the children with ASD, the greater the activity in regions associated with reward processing, including NAc and vmPFC. Results provide new evidence for the social motivation theory (5) by suggesting that the degree of social reward impairment varies as a function of social abilities, supporting a link between being *tuned into* the social world and being *rewarded by* the social world.

6) From what I could tell, no manipulation check was done to measure some aspect of how rewarding the stimuli were to participants. This seems critical if the authors want to make strong reverse inferences back to the social motivation theory.

We thank the reviewer for this comment. While we agree that it would have been optimal to have a behavioral measure of vocal reward for these children, we were not confident that we would be able to elicit valid behavioral responses regarding an abstract concept like “reward” from children with ASD as young as 7-8 years old with moderate to severe communication deficits. Indeed, there is concern that neurotypical children in this age range might have difficulty comprehending the nature of “reward” in the context of their mother’s voice, a ubiquitous sound source in many children’s environment since before birth. Furthermore, to our knowledge, there are no validated behavioral measures for auditory processing analogous to eye-tracking (19, 21, 22) that might have been used for children in this age range to infer reward processing for these vocal sounds.

7) The authors should include a limitations section to their paper.

We have added a paragraph that identifies limitations of the current study to the revised Discussion section.

**References**

1. Association AP. Diagnostic and statistical manual of mental disorders: DSM-5. Washington, D.C.: American Psychiatric Association; 2013.

2. Kjelgaard MM, Tager-Flusberg H. An Investigation of Language Impairment in Autism: Implications for Genetic Subgroups. Lang Cogn Process. 2001;16(2-3):287-308. doi:10.1080/01690960042000058.

3. Tager-Flusberg H, R. Paul, and C. Lord. Language and Communication in Autism. In: F.R. Volkmar RP, and A. Klin, editor. Handbook of Autism and Pervasive Developmental Disorders, Volume 1: Diagnosis, Development, Neurobiology, and Behavior. I. Hoboken, NJ: John Wiley & Sons, Incorporated; 2005. p. 335-64.

4. Lord C, Rutter M, DiLavore PC, Risi S, Gotham K, Bishop S. Autism diagnostic observation schedule, second edition. Torrance, CA: Western Psychological Services; 2012.

5. Chevallier C, Kohls G, Troiani V, Brodkin ES, Schultz RT. The social motivation theory of autism. Trends Cogn Sci. 2012;16(4):231-9. doi:10.1016/j.tics.2012.02.007.

6. Dawson G, Carver L, Meltzoff AN, Panagiotides H, McPartland J, Webb SJ. Neural correlates of face and object recognition in young children with autism spectrum disorder, developmental delay, and typical development. Child Dev. 2002;73(3):700-17. doi:Doi 10.1111/1467-8624.00433.

7. Cohen JR, Asarnow RF, Sabb FW, Bilder RM, Bookheimer SY, Knowlton BJ, Poldrack RA. Decoding continuous variables from neuroimaging data: basic and clinical applications. Front Neurosci. 2011;5:75. doi:10.3389/fnins.2011.00075.

8. Yerys BE, Jankowski KF, Shook D, Rosenberger LR, Barnes KA, Berl MM, Ritzl EK, Vanmeter J, Vaidya CJ, Gaillard WD. The fMRI success rate of children and adolescents: typical development, epilepsy, attention deficit/hyperactivity disorder, and autism spectrum disorders. Hum Brain Mapp. 2009;30(10):3426-35. doi:10.1002/hbm.20767.

9. Jao Keehn RJ, Sanchez SS, Stewart CR, Zhao W, Grenesko-Stevens EL, Keehn B, Muller RA. Impaired downregulation of visual cortex during auditory processing is associated with autism symptomatology in children and adolescents with autism spectrum disorder. Autism Res. 2017;10(1):130-43. doi:10.1002/aur.1636.

10. Wadsworth HM, Maximo JO, Donnelly RJ, Kana RK. Action simulation and mirroring in children with autism spectrum disorders. Behav Brain Res. 2018;341:1-8. doi:10.1016/j.bbr.2017.12.012.

11. Oberwelland E, Schilbach L, Barisic I, Krall SC, Vogeley K, Fink GR, Herpertz-Dahlmann B, Konrad K, Schulte-Ruther M. Young adolescents with autism show abnormal joint attention network: A gaze contingent fMRI study. Neuroimage Clin. 2017;14:112-21. doi:10.1016/j.nicl.2017.01.006.

12. Wadsworth HM, Maximo JO, Lemelman AR, Clayton K, Sivaraman S, Deshpande HD, Ver Hoef L, Kana RK. The Action Imitation network and motor imitation in children and adolescents with autism. Neuroscience. 2017;343:147-56. doi:10.1016/j.neuroscience.2016.12.001.

13. Utzerath C, Schmits IC, Buitelaar J, de Lange FP. Adolescents with autism show typical fMRI repetition suppression, but atypical surprise response. Cortex. 2018;109:25-34.

14. Greene RK, Spanos M, Alderman C, Walsh E, Bizzell J, Mosner MG, Kinard JL, Stuber GD, Chandrasekhar T, Politte LC, Sikich L, Dichter GS. The effects of intranasal oxytocin on reward circuitry responses in children with autism spectrum disorder. J Neurodev Disord. 2018;10(1):12. doi:10.1186/s11689-018-9228-y.

15. Vogan VM, Francis KE, Morgan BR, Smith ML, Taylor MJ. Load matters: neural correlates of verbal working memory in children with autism spectrum disorder. J Neurodev Disord. 2018;10(1):19. doi:10.1186/s11689-018-9236-y.

16. Lynch CJ, Breeden AL, You X, Ludlum R, Gaillard WD, Kenworthy L, Vaidya CJ. Executive Dysfunction in Autism Spectrum Disorder Is Associated With a Failure to Modulate Frontoparietal-insular Hub Architecture. Biological Psychiatry: Cognitive Neuroscience and Neuroimaging. 2017;2(6):537-45.

17. Button KS, Ioannidis JP, Mokrysz C, Nosek BA, Flint J, Robinson ES, Munafo MR. Power failure: why small sample size undermines the reliability of neuroscience. Nat Rev Neurosci. 2013;14(5):365-76. doi:10.1038/nrn3475.

18. Nee DE. Correspondence: fMRI replicability depends upon sufficient individual level data. bioRxiv. 2018.

19. Safra L, Ioannou C, Amsellem F, Delorme R, Chevallier C. Distinct effects of social motivation on face evaluations in adolescents with and without autism. Sci Rep. 2018;8(1):10648. doi:10.1038/s41598-018-28514-7.

20. Chevallier C, Tonge N, Safra L, Kahn D, Kohls G, Miller J, Schultz RT. Measuring Social Motivation Using Signal Detection and Reward Responsiveness. PLoS One. 2016;11(12):e0167024. doi:10.1371/journal.pone.0167024.

21. Dubey I, Ropar D, de CHAF. Brief Report: A Comparison of the Preference for Viewing Social and Non-social Movies in Typical and Autistic Adolescents. J Autism Dev Disord. 2017;47(2):514-9. doi:10.1007/s10803-016-2974-3.

22. Dubey I, Ropar D, Hamilton AF. Measuring the value of social engagement in adults with and without autism. Mol Autism. 2015;6:35. doi:10.1186/s13229-015-0031-2.

23. Sepeta L, Tsuchiya N, Davies MS, Sigman M, Bookheimer SY, Dapretto M. Abnormal social reward processing in autism as indexed by pupillary responses to happy faces. J Neurodev Disord. 2012;4(1):17. doi:10.1186/1866-1955-4-17.

24. Wechsler D. The Wechsler Abbreviated Scale of Intelligence. San Antonio, TX: The Psychological Corporation; 1999.

25. Wagner RK, Torgesen JK, Rashotte CA. Comprehensive Test of Phonological Processing (CTOPP). Pro-Ed I, editor. Austin, TX1999.

26. Semel E, Wiig EH, Secord WH. Clinical evaluation of language fundamentals – Fourth edition (CELF-4). San Antonio, TX: Psychological Corporation; 2003.

27. Belin P, Zatorre RJ, Lafaille P, Ahad P, Pike B. Voice-selective areas in human auditory cortex. Nature. 2000;403(6767):309-12. doi:10.1038/35002078.

28. Gotham K, Pickles A, Lord C. Standardizing ADOS scores for a measure of severity in autism spectrum disorders. J Autism Dev Disord. 2009;39(5):693-705. doi:10.1007/s10803-008-0674-3.

29. Hus V, Gotham K, Lord C. Standardizing ADOS domain scores: separating severity of social affect and restricted and repetitive behaviors. J Autism Dev Disord. 2014;44(10):2400-12. doi:10.1007/s10803-012-1719-1.

30. Abrams DA, Chen T, Odriozola P, Cheng KM, Baker AE, Padmanabhan A, Ryali S, Kochalka J, Feinstein C, Menon V. Neural circuits underlying mother's voice perception predict social communication abilities in children. Proc Natl Acad Sci U S A. 2016;113(22):6295-300. doi:10.1073/pnas.1602948113.

31. Horikawa T, Kamitani Y. Generic decoding of seen and imagined objects using hierarchical visual features. Nat Commun. 2017;8:15037. doi:10.1038/ncomms15037.

32. Kashyap S, Ivanov D, Havlicek M, Sengupta S, Poser BA, Uludag K. Resolving laminar activation in human V1 using ultra-high spatial resolution fMRI at 7T. Sci Rep. 2018;8(1):17063. doi:10.1038/s41598-018-35333-3.

33. Wen H, Shi J, Chen W, Liu Z. Deep Residual Network Predicts Cortical Representation and Organization of Visual Features for Rapid Categorization. Sci Rep. 2018;8(1):3752. doi:10.1038/s41598-018-22160-9.

34. DeCasper AJ, Fifer WP. Of human bonding: newborns prefer their mothers' voices. Science. 1980;208(4448):1174-6.

35. Seltzer LJ, Prososki AR, Ziegler TE, Pollak SD. Instant messages vs. speech: hormones and why we still need to hear each other. Evolution and Human Behavior. 2012;33(1):42-5. doi:10.1016/j.evolhumbehav.2011.05.004.

36. Seltzer LJ, Ziegler TE, Pollak SD. Social vocalizations can release oxytocin in humans. Proc Biol Sci. 2010;277(1694):2661-6. doi:10.1098/rspb.2010.0567.

37. Thoman EB, Korner AF, Beasonwilliams L. Modification of Responsiveness to Maternal Vocalization in Neonate. Child Dev. 1977;48(2):563-9. doi:DOI 10.1111/j.1467-8624.1977.tb01198.x.

38. Lamb ME. Developing trust and perceived effectance in infancy. Advances in Infancy Research. Norwood, NJ: Ablex; 1981. p. 101-27.

39. Klin A. Young autistic children's listening preferences in regard to speech: a possible characterization of the symptom of social withdrawal. J Autism Dev Disord. 1991;21(1):29-42.

40. Kuhl PK, Coffey-Corina S, Padden D, Dawson G. Links between social and linguistic processing of speech in preschool children with autism: behavioral and electrophysiological measures. Dev Sci. 2005;8(1):F1-F12. doi:10.1111/j.1467-7687.2004.00384.x.

41. Abrams DA, Lynch CJ, Cheng KM, Phillips J, Supekar K, Ryali S, Uddin LQ, Menon V. Underconnectivity between voice-selective cortex and reward circuitry in children with autism. Proc Natl Acad Sci U S A. 2013;110(29):12060-5. doi:10.1073/pnas.1302982110.

42. Mitchell TM, Shinkareva SV, Carlson A, Chang KM, Malave VL, Mason RA, Just MA. Predicting human brain activity associated with the meanings of nouns. Science. 2008;320(5880):1191-5. doi:10.1126/science.1152876.

43. Falk EB, Berkman ET, Mann T, Harrison B, Lieberman MD. Predicting persuasion-induced behavior change from the brain. J Neurosci. 2010;30(25):8421-4. doi:10.1523/JNEUROSCI.0063-10.2010.

44. Hoeft F, McCandliss BD, Black JM, Gantman A, Zakerani N, Hulme C, Lyytinen H, Whitfield-Gabrieli S, Glover GH, Reiss AL, Gabrieli JD. Neural systems predicting long-term outcome in dyslexia. Proc Natl Acad Sci U S A. 2011;108(1):361-6. doi:10.1073/pnas.1008950108.

45. Dosenbach NU, Nardos B, Cohen AL, Fair DA, Power JD, Church JA, Nelson SM, Wig GS, Vogel AC, Lessov-Schlaggar CN, Barnes KA, Dubis JW, Feczko E, Coalson RS, Pruett JR, Jr., Barch DM, Petersen SE, Schlaggar BL. Prediction of individual brain maturity using fMRI. Science. 2010;329(5997):1358-61. doi:10.1126/science.1194144.

46. Lohmann G, Erfurth K, Muller K, Turner R. Critical comments on dynamic causal modelling. Neuroimage. 2012;59(3):2322-9. doi:10.1016/j.neuroimage.2011.09.025.

47. Floris DL, Lai MC, Auer T, Lombardo MV, Ecker C, Chakrabarti B, Wheelwright SJ, Bullmore ET, Murphy DG, Baron-Cohen S, Suckling J. Atypically rightward cerebral asymmetry in male adults with autism stratifies individuals with and without language delay. Hum Brain Mapp. 2016;37(1):230-53. doi:10.1002/hbm.23023.

48. Lombardo MV, Pierce K, Eyler LT, Carter Barnes C, Ahrens-Barbeau C, Solso S, Campbell K, Courchesne E. Different functional neural substrates for good and poor language outcome in autism. Neuron. 2015;86(2):567-77. doi:10.1016/j.neuron.2015.03.023.

49. Hong SJ, Valk SL, Di Martino A, Milham MP, Bernhardt BC. Multidimensional Neuroanatomical Subtyping of Autism Spectrum Disorder. Cereb Cortex. 2018;28(10):3578-88. doi:10.1093/cercor/bhx229.

50. Pantelis PC, Byrge L, Tyszka JM, Adolphs R, Kennedy DP. A specific hypoactivation of right temporo-parietal junction/posterior superior temporal sulcus in response to socially awkward situations in autism. Soc Cogn Affect Neurosci. 2015;10(10):1348-56. doi:10.1093/scan/nsv021.

51. McLaren DG, Ries ML, Xu G, Johnson SC. A generalized form of context-dependent psychophysiological interactions (gPPI): a comparison to standard approaches. Neuroimage. 2012;61(4):1277-86. doi:10.1016/j.neuroimage.2012.03.068.

52. Jones CR, Happe F, Baird G, Simonoff E, Marsden AJ, Tregay J, Phillips RJ, Goswami U, Thomson JM, Charman T. Auditory discrimination and auditory sensory behaviours in autism spectrum disorders. Neuropsychologia. 2009;47(13):2850-8. doi:10.1016/j.neuropsychologia.2009.06.015.

53. Bonnel A, Mottron L, Peretz I, Trudel M, Gallun E, Bonnel AM. Enhanced pitch sensitivity in individuals with autism: a signal detection analysis. J Cogn Neurosci. 2003;15(2):226-35. doi:10.1162/089892903321208169.

54. Heaton P, Williams K, Cummins O, Happe F. Autism and pitch processing splinter skills: a group and subgroup analysis. Autism. 2008;12(2):203-19. doi:10.1177/1362361307085270.

55. Heaton P. Interval and contour processing in autism. J Autism Dev Disord. 2005;35(6):787-93. doi:10.1007/s10803-005-0024-7.

56. Bartolucci G, Pierce S, Streiner D, Eppel PT. Phonological investigation of verbal autistic and mentally retarded subjects. J Autism Child Schizophr. 1976;6(4):303-16.

57. Bartolucci G, Pierce SJ. A preliminary comparison of phonological development in autistic, normal, and mentally retarded subjects. Br J Disord Commun. 1977;12(2):137-47.

58. Bishop DV, Maybery M, Wong D, Maley A, Hill W, Hallmayer J. Are phonological processing deficits part of the broad autism phenotype? Am J Med Genet B Neuropsychiatr Genet. 2004;128B(1):54-60. doi:10.1002/ajmg.b.30039.

59. Poldrack RA, Baker CI, Durnez J, Gorgolewski KJ, Matthews PM, Munafo MR, Nichols TE, Poline JB, Vul E, Yarkoni T. Scanning the horizon: towards transparent and reproducible neuroimaging research. Nat Rev Neurosci. 2017;18(2):115-26. doi:10.1038/nrn.2016.167.

[Editors' note: further revisions were suggested prior to acceptance, as described below.]

We appreciate the enthusiasm of reviewer #1 for our revised manuscript. To further address the Reviewing Editor’s and reviewer #3’s concern regarding our sample size, we have included an additional sentence to the revised limitations section, which is tracked in the revised manuscript. We have also modified the scatterplots as requested by reviewer #3.